# UIT-Pred: Universal Intermittent Trajectory Predictor for Autonomous Driving

## Abstract

Trajectory prediction is a fundamental component of autonomous driving, requiring models that can handle intermittent observation patterns such as variable-length histories and missing data. Existing state-of-the-art methods, however, often assume fixed-length trajectories and complete input, which challenges their applicability in real-world scenarios where sensor occlusions, communication delays, and temporal sparsity are common. Moreover, conventional approaches typically address tasks such as trajectory prediction, variable-length modeling, or missing data handling in isolation, making them less effective in multi-task settings that naturally arise in practice. To address these challenges, we propose Universal Intermittent Trajectory Predictor (UIT-Pred) that processes inputs with the time index features, which capture temporal variations to effectively adapt to diverse input patterns within the domain. Particularly, we extend recent State Space Models (SSMs) by introducing the Bidirectional Time Decay Mamba (BTD-Mamba), designed to capture dependencies both forward and backward along the sequence. By integrating a decay process, BTD-Mamba effectively analyzes trajectories while maintaining relationships under intermittent observation. Furthermore, the proposed prediction module employs state encoding to capture the underlying motion patterns in the input data and models a multimodal trajectory distribution to account for uncertainty in future predictions. These components are fused through a unified fusion module, enabling the model to jointly reason over observed dynamics and potential future behaviors. Extensive experiments on Argoverse 1 and Argoverse 2 datasets validate the effectiveness of the proposed model. By simultaneously handling prediction, variable-length observations, and missing inputs within a universal architecture, the framework proposes to meet the challenges of real-world autonomous driving systems.

## 1 Introduction

Trajectory prediction is a core challenge in autonomous driving, as safe navigation requires anticipating the future behaviors of surrounding agents under uncertain and dynamic conditions. While deep learning models Huang et al. (2025); Karim et al. (2024) has enabled significant progress, they are designed for fixed-length trajectories and complete observations. In practice, however, observations are often intermittent: variable-length sequences arise when agents enter or exit the sensor's field of view at different times or are observed for varying durations, while missing data occurs due to sensor occlusions or communication delays. This discrepancy in the observation can degrade the performance of state-of-the-art methods unless the model explicitly handles these issues Xu & Fu (2024); Qiu et al. (2025).

Some of the approaches address the issue of variable length trajectories. Xu & Fu (2024) attribute length bias in Transformers to positional encoding and layer normalization, proposing specialized subnetworks for different sequence lengths. Li et al. (2024b) introduce a length-agnostic knowledge distillation (LaKD) module that dynamically transfers knowledge across trajectories. Qiu et al. (2025) proposes Contrastive Learning for Length Shift (CLLS), which uses contrastive learning during training to help the model learn length-invariant features and reduce the effect of varying observation lengths. Although these approaches show some effectiveness, they rely on generating multiple augmented versions of each trajectory sequence, which expands the input space and increases the overall complexity of the training.

Other methods like TranSPORTmer Capellera et al. (2024) and MS-TIP Chib et al. (2024) both apply masking techniques within transformer frameworks to address missing data, with TranSPORTmer applied to sports scenarios and MS-TIP designed for pedestrian trajectory recovery. In contrast, U2Diff Capellera et al. (2025) simultaneously reconstructs missing agent states and estimates uncertainty, focusing on the sports domain. However, these approaches depend on masking strategies to reconstruct missing observations, but this adds complexity in handling masked vs. unmasked inputs. Although effective in the sports and pedestrian domains, their applicability to autonomous driving scenarios remains largely unexplored. Additionally, State-of-the-art methods usually address prediction, variable-length observation, or missing input separately, overlooking the multi-task nature of real-world systems. Since diverse scenarios might happen in real practice, it is essential to develop a unified approach that can handle various input conditions, as illustrated in Figure. 1.

To bridge the aforementioned challenges, we propose the Universal Intermittent Trajectory Predictor (UIT-Pred), a unified architecture designed to effectively handle varying input conditions in trajectory forecasting. UIT-Pred transforms diverse input formats into a generalizable schema through time-aware input representations. Specifically, we derive two complementary temporal features from time indices: scaled timestamps to account for varying time ranges, and inter-observation interval features to capture the timing gaps between observations. These temporal cues enable the model to capture motion dynamics without depending on fixed time references or explicit validity masks. Building on the capabilities of recent State Space Models (SSMs), particularly the Mamba architecture, we introduce an enhanced Bidirectional Time Decay Mamba (BTD-Mamba) module, which captures sequential dependencies in both forward and backward directions across input observations. Additionally, a decay mechanism is incorporated to maintain the continuity and integrity of temporal relationships despite intermittent observations. Furthermore, in the proposed prediction module, we introduce a learnable state embedding to effectively capture the underlying dynamics of variable-length input sequences and missing observations.

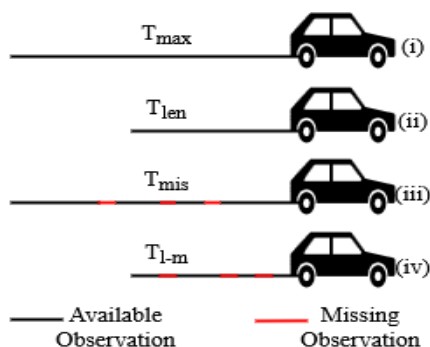

Figure 1: The following input conditions reflect scenarios commonly encountered in real-world traffic systems: (i) full observation is available; (ii) observations are available, but of variable lengths; (iii) some observation points are missing; and (iv) both variable-length and missing observations occur simultaneously.

vations. This embedding provides a compact yet informative representation of the agent's motion history, maintaining temporal continuity and capturing key behavioral patterns. To further enrich this representation, we employ a cross-attention mechanism to integrate global context, including nearby agents and road topology. Finally, the enhanced state embedding is fused with the agent's multimodal features through the proposed unified fusion module, enabling mutual learning and enhancing prediction accuracy.

Our contributions are summarized as follows: **(i)** We propose UIT-Pred, a generalizable architecture that effectively handles diverse input conditions including variable-length histories and missing input data in trajectory forecasting. **(ii)** We extend the Mamba architecture by introducing the Bidirectional Time Decay Mamba (BTD-Mamba) module, designed to extract rich spatiotemporal features from diverse forms of intermittent trajectory inputs. **(iii)** We introduce a novel prediction module that generates a learnable state embedding to capture the dynamics of observed motion patterns, which is then fused with the multimodal output to enhance trajectory prediction. **(iv)** Extensive experiments on the Argoverse 1 and Argoverse 2 benchmarks demonstrate the consistent and strong performance of our method.

## 2 RELATED WORK

### 2.1 TRAJECTORY PREDICTION

Trajectory prediction is the task of forecasting the future paths of moving agents, such as vehicles, pedestrians, or cyclists, based on a sufficiently long, fixed-length history of observed positions.

In recent years, numerous approaches have been developed to address this challenge Chen et al. (2025a), Messaoud et al. (2025). To model interactions between agents and the map, graph neural networks Wang et al. (2025a); Chen et al. (2025b) and attention-based mechanisms Xin et al. (2025); Bharilya et al. (2025); Lee et al. (2024); Huang et al. (2025) have been widely employed. Furthermore, to capture the inherent uncertainty of road agents, researchers generate multimodal predictions using GANs Wang et al. (2025b), flow-based models Liang et al. (2023), and diffusion models Capellera et al. (2025), Wang et al. (2024b), Neumeier et al. (2024). Additionally, goal-based approaches Afshar et al. (2024), Xing et al. (2025) have gained traction, where multi-modal goals are first generated through sampling or learning, followed by trajectory prediction conditioned on these goals.

Recently, the Mamba Gu & Dao (2023) framework has revived interest in state space models (SSMs) as promising alternatives to Transformers Vaswani et al. (2017), owing to their ability to reduce attention complexity and capture long-term dependencies. Mamba has shown strong potential across diverse domains, including natural language processing Zhao et al. (2025); Wang et al. (2024a) and computer vision Hatamizadeh & Kautz (2025); Yu & Wang (2025). Building on these advances, our method integrates the strengths of Mamba with Transformer architectures to achieve superior performance in the unified trajectory prediction task.

## 2.2 Trajectory Prediction for Length Shift

Trajectory prediction with variable observation lengths has received growing attention in recent years. Xu & Fu (2024) attribute length bias in Transformers to positional encoding and layer normalization, proposing specialized subnetworks for different sequence lengths. Li et al. (2024b) introduce a length-agnostic knowledge distillation (LaKD) module that dynamically transfers knowledge across trajectories. Qiu et al. (2025) proposes Contrastive Learning for Length Shift (CLLS), which uses contrastive learning during training to help the model learn length-invariant features and reduce the effect of varying observation lengths. Additionally, methods like ITPNet Li et al. (2024a), MOE Sun et al. (2022), DTO Monti et al. (2022), and SingularTrajectory Bae et al. (2024) perform instantaneous trajectory prediction by forecasting future motion based on a very short history, typically the last two time steps, but they depend on a fixed input length. In contrast, our proposed method handles variable-length observations.

## 2.3 Trajectory Imputation

Trajectory imputation aims to reconstruct unobserved agent states by leveraging contextual and historical motion data. Earlier work on time-series imputation has explored autoregressive RNNs for filling in missing values Cao et al. (2018). GC-VRNN Xu et al. (2023) couples a variational RNN with a spatio-temporal GNN to reconstruct missing points and forecast futures in one framework. Recently, TranSPORTmer Capellera et al. (2024) applied input masking within a transformer architecture to impute missing observations, outperforming task-specific baselines in both player and ball tracking. Similarly, MS-TIP Chib et al. (2024) employed diagonal masked self-attention in transformers to recover missing data in pedestrian trajectories. U2Diff Capellera et al. (2025) introduced a unified diffusion-based model that reconstructs missing agent states while estimating state-wise uncertainty. While these methods focus on imputation, their primary applications are in sports or pedestrian settings. The challenge of handling missing data in the context of autonomous driving remains largely underexplored.

## 3 Proposed Method

**Problem Definition** In autonomous driving trajectory prediction, the goal is to forecast a target agent's future motion based on past observations and contextual information (e.g., maps and surrounding agents). Real-world data often contain intermittent patterns, including variable-length observations and missing values. Formally, for each agent, the observed sequence is denoted as $\tilde{X} = \{\tilde{x}_1, \tilde{x}_2, \ldots, \tilde{x}_{t_{obs}}\}$, where each $\tilde{x}_i$ contains coordinates and velocity; the length $t_{obs}$ varies and some $\tilde{x}_i$ may be missing. The task is to predict the future trajectory $\hat{Y} = \{\hat{y}_1, \ldots, \hat{y}_{t_{pred}}\}$, where each $\hat{y}_j$ represents agent positions over a horizon $t_{pred}$, using observations $\tilde{X}$ and context. Here, $t_{total} = t_{max} + t_{pred}$ denotes the full sequence length, with $t_{max}$ as the agent's history length.

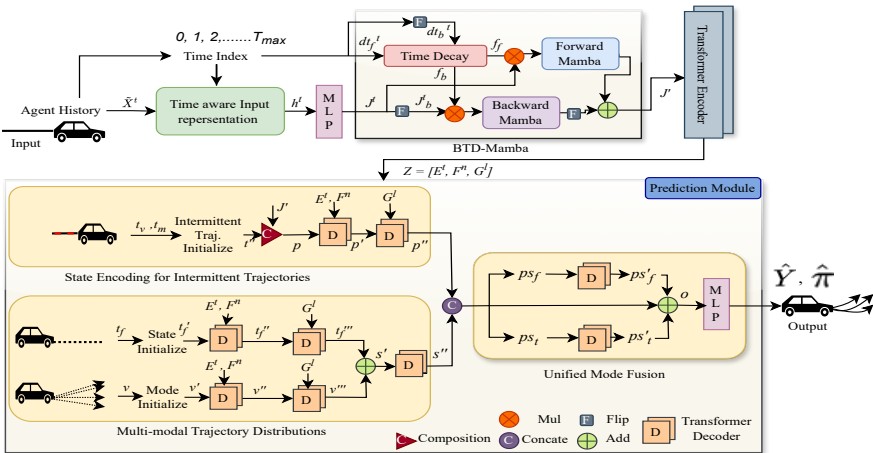

Figure 2: Illustration of our UIT-Pred framework. For simplicity, the neighbor and map encoding branches are omitted. Here, $F^n$ represents the encoded features of neighboring agents, and $G^l$ denotes the encoded lane information.

## 3.1 TIME-AWARE INPUT REPRESENTATION

We represent each trajectory as a sequence of temporal states with spatial and contextual features. For a given target agent, the input at each timestep $i$ consists of concatenated state features including spatial coordinates $x_t^{(i)}$ and velocity $\text{vel}_t^{(i)}$ to form a comprehensive feature vector $\tilde{\mathbf{X}}_i^{\mathbf{t}} = \left[ x_t^{(i)} \parallel \text{vel}_t^{(i)} \right]$ where $\parallel$ denotes concatenation. To handle varying input lengths and missing observations, we enhance the agent's feature state by incorporating two complementary temporal features. First, we compute a *scaled timestamps*,

$$t_i = 1 - \frac{\tau_i - \tau_{\min}}{\tau_{\max} - \tau_{\min}}, \quad t_i \in [0, 1] \tag{1}$$

where $\tau_i$ is the original absolute timestamp (e.g., $\tau_{\mathbf{i}} = [0, 1, \ldots, t_{\text{obs}}]$). This maps each time index to the range $[0, 1]$, ensuring the model to learn temporal patterns in the observations without bias toward any specific sequence length. Second, we compute the *inter-observation interval feature*, which represents the time elapsed since the previous valid observation,

$$\Delta t_i = \tau_i - \tau_{i-1}, \quad i > 1, \quad \Delta t_1 = 0 \tag{2}$$

The feature $\Delta t_i$ allow the model to reason about missing observations and We explicitly tell the model how much time has passed since the previous observation., without explicitly relying on binary validity masks. Thus, the final input at each timestep is represented as,

$$\mathbf{h}_i^t = \left[ x_t^{(i)} \parallel \text{vel}_{\text{t}}^{(i)} \parallel t_{\text{norm}}^{(i)} \parallel \Delta t^{(i)} \right]_{i=1}^{t_{\text{obs}}} \tag{3}$$

This formulation integrates spatial-temporal states, scaled timestamps to handle varying time ranges, and inter-observation interval feature to represent the timing differences in the observation.

For each neighboring agent, the input representation is constructed similarly to that of the target agent, using all available timestamps within a fixed observation window to form the feature vector $\mathbf{h}_i^n$. For lane segment points, the input $\mathbf{h}_i^l$ combines geometric and visibility information, following the design in Zhang et al. (2024). (refer to Appendix A.1 for further details).

## 3.2 BIDIRECTIONAL TIME-DECAY MAMBA (BTD-MAMBA)

Mamba blocks inherently support variable-length sequences through recurrent state-space updates. Building on this, we introduce BTD-Mamba, as shown in Figure. 2, an enhanced state-space model

designed to handle both variable-length and missing trajectory data. It extends Mamba by incorporating a time decay mechanism that modulates hidden states based on inter-observation intervals. Initially, we compute forward and backward inter-arrival times as,

$$\mathbf{dt}_f^t = [\Delta t_0, \Delta t_1, \ldots, \Delta t_{T_{\text{obs}}-1}], \quad \mathbf{dt}_b^t = \text{flip}(\mathbf{dt}_f^t), \quad \mathbf{dt}^{(f,b)} = [\mathbf{dt}_f^t, \mathbf{dt}_b^t] \tag{4}$$

The concatenated inter-arrival times $\mathbf{dt}^{(f,b)}$ encode both forward and backward time gaps between observations. We then project this into a scaling feature space using,

$$[f_{\text{f}}, f_{\text{b}}]_{exp} = \frac{1}{\exp\left(\text{ReLU}\left(\phi_s(\mathbf{dt}^{(f,b)}; W_s)\right)\right)} \tag{5}$$

Here, $\phi_s(\cdot)$ is a projection function parameterized by weights $W_s$, implemented as a multi-layer perceptron (MLP) with ReLU activation. Equation 4 is designed to calculate the distance from the last observation to the current time step, which helps quantify the influence of temporal gaps, particularly when dealing with complex missing patterns. The key insight is that the influence of a variable that has been missing for a period decreases over time. Therefore, in Equation 5 we utilize a negative exponential function combined with ReLU to ensure that the influence decays monotonically within a reasonable range between 0 and 1. Moreover, we apply Mamba bidirectionally with a time-decay mechanism, effectively capturing irregular temporal intervals and modeling temporal gaps by leveraging inter-arrival times in both directions. The embedded sequence $\mathbf{J^t} = \{\mathbf{j}_1, \mathbf{j}_2, \ldots, \mathbf{j}_{t_{\text{obs}}}\}$, where each $\mathbf{j}_i$ is generated by passing $\mathbf{h}_i^t$ through an MLP, is processed both in its original order and reversed order $\mathbf{J}_{\text{b}}^{\bar{t}} = \{\mathbf{j}_{t_{\text{obs}}}, \ldots, \mathbf{j}_2, \mathbf{j}_1\}$ by the revised Mamba block as described below,

$$J' = \left(J^t \odot \text{f}_f\right) * C_{\text{forw}} + \text{Flip}\left(\left(J_b^{\bar{t}} \odot \text{f}_{\text{b}}\right) * C_{\text{back}}\right) \tag{6}$$

where $\odot$ is element-wise multiplication, $*$ is convolution, $\text{Flip}(\cdot)$ reverses the sequence, and $C_{\text{forw}}$, $C_{\text{back}}$ are learnable convolution kernels for forward and backward directions, respectively.

**Interaction Representation**   The neighboring agent features $\mathbf{h}_i^n$ are first embedded using an MLP to produce $\mathbf{J^n} \in \mathbb{R}^{N_a \times d}$. These embeddings are then passed through Mamba blocks and subsequently refined via residual layers with skip connections and normalization, as expressed below,

$$\mathbf{J^n} = \text{MLP}(\mathbf{h}_i^n), \quad \mathbf{F} = \text{MambaBlocks}(\mathbf{J^n}), \quad \mathbf{F}' = \text{Norm}(\mathbf{F} + \mathbf{J^n}) \tag{7}$$

Lane features $\mathbf{h}^l$ are encoded via a PointNet-based encoder Zhang et al. (2024), yielding map embeddings $\mathbf{G}' = \text{PNEncoder}(\mathbf{h}^l) \in \mathbb{R}^{N_l \times d}$. Interactions among the target agent, neighbors, and map are captured by concatenating their embeddings and passing them through a Transformer encoder.

$$\mathbf{I} = [\mathbf{J}' \parallel \mathbf{F}' \parallel \mathbf{G}'], \quad \mathbf{Z} = \text{TransformerEncoder}(\mathbf{I}, \mathbf{I}, \mathbf{I}) \tag{8}$$

where $Z = [E^t \in \mathbb{R}^{1 \times d}, F^n \in \mathbb{R}^{N_a \times d}, G^l \in \mathbb{R}^{N_l \times d}] \in \mathbb{R}^{(1+N_a+N_l) \times d}$ captures joint contextual representations for predicting the next trajectory. Here, $N_a$ denotes the number of neighboring agents, $N_l$ denotes the number of lane agents, and $d$ is the embedding dimension.

## 3.3   PREDICTION MODULE

**State Encoding for Intermittent Trajectories**   In our framework, to effectively capture the state and temporal dynamics of trajectories amid variable sequence lengths and missing observations, we introduce a *Variable-Step Temporal Representation (VST)* that accounts for unobserved, varying sequence lengths, alongside a *Missing-Step Temporal Embedding (MST)* to handle missing data. For a sequence of length $t_{\text{len}}$, we construct a time vector from $t_{\text{max}}$ down to $t_{\text{max}} - t_{\text{len}} + 1$. The vector $t_v \in \mathbb{Z}^{t_{uv}}$ represents the variable-length unobserved time steps, with $t_{uv} = t_{\text{max}} - t_{\text{len}}$, while $t_m$ denotes the number of missing steps $m_i$,

$$\mathbf{t}_v = [t_{\text{max}}, \ldots, t_{\text{max}} - t_{\text{len}} + 1], \quad \mathbf{t}_m = [m_1, \ldots m_{t_m}], \quad \mathbf{t} = [t_v, t_m] \tag{9}$$

These $\mathbf{t}$ are normalized, scaled, and passed through an MLP to generate learnable temporal features,

$$\mathbf{t}_{\text{scaled}} = 0.1 \times \mathbf{t} + 0.1, \quad \mathbf{t}' = \text{MLP}(\mathbf{t}_{\text{scaled}}) \in \mathbb{R}^{(t_m + t_{uv}) \times d} \tag{10}$$

The resulting embeddings $\mathbf{t}'$ are added to repeated latent features of the target agent, enriched with state information via a GRU,

$$\mathbf{t}'' = \mathbf{t}' \oplus \text{GRU}(\mathbf{E^t}) \in \mathbb{R}^{(t_m + t_{uv}) \times d} \tag{11}$$

where $\oplus$ denotes broadcasting addition across time, and $\mathbf{t}'' = [t'_v, t'_m]$ contains the final VST $v'_t \in \mathbb{R}^{t_{uv} \times d}$ and MST $m'_t \in \mathbb{R}^{t_m \times d}$ embeddings. Moreover, we integrate the observed sequence features with the VST and MST embedding to create a unified observed sequence, enabling seamless learning across all components. Initially, a ternary mask $\mathbf{M} \in \{0, 1, 2\}^{t_{\max}}$ is constructed,

$$\mathbf{M}_{i,t} = \begin{cases} 0 & \text{if time step } t \text{ is observed,} \\ 1 & \text{if time step } t \in \mathbf{t}'_v, \\ 2 & \text{if time step } t \in \mathbf{t}'_m. \end{cases} \quad (12)$$

Using this mask, we form the tensor $\mathbf{p} \in \mathbb{R}^{t_{\max} \times d}$, which combines observed data, the final VST embedding, and the MST embedding, ensuring each time step is assigned the correct representation,

$$\mathbf{p}[i,t] = \begin{cases} \mathbf{J}'[i,j], & \text{if } \mathbf{M}[i,t] = 0 \\ \mathbf{t}'_m[i,k], & \text{if } \mathbf{M}[i,t] = 1 \\ \mathbf{t}'_{\text{imp}}[i,l], & \text{if } \mathbf{M}[i,t] = 2 \end{cases} \quad (13)$$

where $j, k, l$ index into the observed data, VST embeddings, and MST embeddings respectively. Finally, the reconstructed full past-length sequence $\mathbf{p} \in \mathbb{R}^{t_{\max} \times d}$ is generated, enriched with agent information $A^{tn} = [E^t, F^n] \in \mathbb{R}^{N_a + 1 \times d}$ and lane information $G^l \in \mathbb{R}^{N_l \times d}$ through cross-attention,

$$\mathbf{p}' = \mathcal{D}(\mathbf{p}, A^{tn}, A^{tn}) \in \mathbb{R}^{t_{max} \times d}, \quad \mathbf{p}'' = \mathcal{D}(\mathbf{p}', G^l, G^l) \in \mathbb{R}^{t_{max} \times d} \quad (14)$$

where $\mathcal{D}$ is the Transformer decoder for cross-attention, and $\mathbf{p}''$ encodes agent and lane information.

**multi-modal trajectory distributions**   To generate a multimodal future trajectory for an agent, we require both mode and state information. The initial mode vector $\mathbf{v} \in \mathbb{R}^{k \times d}$ is embedded and combined with the target agent's encoding to form $\mathbf{v}' \in \mathbb{R}^{k \times d}$, which is then refined via cross-attention with agent $A^{tn} \in \mathbb{R}^{N_a + 1 \times d}$ and lane $G^l \in \mathbb{R}^{N_l \times d}$ information,

$$\mathbf{v}' = \mathbf{v} + \mathbf{E^t}, \quad \mathbf{v}'' = \mathcal{D}(\mathbf{v}', A^{tn}, A^{tn}), \quad \mathbf{v}''' = \mathcal{D}(\mathbf{v}', G^l, G^l) \quad (15)$$

where $\mathbf{v}''' \in \mathbb{R}^{k \times d}$ represents the final mode vector. To generate the state vector $\mathbf{t}_f \in \mathbb{Z}^{t_{pred}}$, a normalized time embedding is constructed and fused with the GRU hidden states,

$$\mathbf{t}'_f = \text{MLP}(0.1 \cdot t + 0.1) \oplus \text{GRU}[E^t] \in \mathbb{R}^{t_{pred} \times d}, \quad t = 1, \ldots, t_{pred} \quad (16)$$

The $t'_f$-enriched states are refined via cross-attention with agent $A^{tn}$ and lane $G^l$ information,

$$\mathbf{t}''_f = \mathcal{D}(t_f, A^{tn}, A^{tn}), \quad \mathbf{t}'''_f = \mathcal{D}(\mathbf{t}''_f, G^l, G^l) \quad (17)$$

The final $\mathbf{t}'''_f$ integrates both agent and lane information. Furthermore, the embedding of multimodal future trajectory $\mathbf{s}' \in \mathbb{R}^{t_{pred} \times k \times d}$ is generated by combining the mode vector $\mathbf{v}'''$ and state vectors $\mathbf{t}'''_f$ and refined with $Z \in \mathbb{R}^{(1 + N_a + N_l) \times d}$,

$$\mathbf{s}' = \mathbf{v}''' \oplus \mathbf{t}'''_f, \quad \mathbf{s}'' = \mathcal{D}(\mathbf{s}, Z, Z) \quad (18)$$

where $\mathbf{s}'' \in \mathbb{R}^{t_{pred} \times k \times d}$ is the refined future trajectory embedding after processing.

**Unified Mode Fusion for Prediction**   To capture relationships between past and future behaviors, we fuse representations $p'' \in \mathbb{R}^{(k \times t_{max} \times d)}$ (after repeating along $k$) and $s'' \in \mathbb{R}^{(k \times t_{pred} \times d)}$ into $ps_f \in \mathbb{R}^{(k \times t_{max} + k \times t_{pred}) \times d}$ by concatenating along the feature dimension $\|_{\text{feat}}$ (preserving temporal resolution) and into $ps_t \in \mathbb{R}^{k \times t_{total} \times d}$ along the temporal dimension $\|_{\text{time}}$ (aligning sequences),

$$ps_f = [p'' \|_{\text{feat}} s''], \quad ps_t = [p'' \|_{\text{time}} s''] \quad (19)$$

These fused mode representations $ps_f$ and $ps_t$ are subsequently processed using cross-attention mechanisms, producing the updated representations $ps'_f$ and $ps'_t$,

$$ps'_f = \mathcal{D}(ps_f, Z, Z) \in \mathbb{R}^{(k \times t_{max} + k \times t_{pred}) \times d}, \quad ps'_t = \mathcal{D}(ps_t, Z, Z) \in \mathbb{R}^{k \times t_{total} \times d} \quad (20)$$

Finally, the outputs from the different blocks are summarized,

$$\mathbf{o} = ps_t \oplus \text{reshape}_{time}(ps'_f) \oplus ps'_t \in \mathbb{R}^{k \times t_{total} \times d} \quad (21)$$

The tensor $\mathbf{o}[-t_{\text{pred}} :]$ is used for downstream multimodal trajectory prediction, while $\mathbf{o}[t_v, t_m]$ is used to predict the corresponding observations,

$$\hat{Y}, \hat{\pi} = MLP(\text{output}[-t_{pred} :]), \quad \hat{x_v}, \hat{x_m} = MLP(\text{output}[t_v, t_m]), \quad (22)$$

where $\hat{Y} \in \mathbb{R}^{k \times t_{pred} \times 2}$ denotes the predicted future trajectories, $\hat{\pi} \in \mathbb{R}^k$ represents the associated mode probabilities, and $\hat{x}_v \in \mathbb{R}^{t_{uv} \times 2}$, $\hat{x}_m \in \mathbb{R}^{t_m \times 2}$ are the reconstructed variable-length steps and missing observations, respectively.

**Model Training.** To supervise the predicted trajectory and its confidence, we employ the Huber loss for trajectory regression, denoted as $L_{\text{reg}}$, along with the cross-entropy loss for confidence classification, denoted as $L_{\text{cls}}$. Additionally, the VST embeddings and MST embedding are supervised using the $L_{\text{u-rg}}$ and $L_{\text{m-rg}}$ losses, respectively, both of which employ the same regression criterion. Furthermore, an endpoint loss $L_{\text{et}}$ is incorporated for all agents, applying the same regression loss function to improve endpoint accuracy. The model is trained end-to-end by combining all the losses,

$$L_{\text{total}} = w_{\text{reg}}L_{\text{reg}} + w_{\text{cls}}L_{\text{cls}} + w_u L_{u\text{-reg}} + w_m L_{m\text{-reg}} + w_e L_{\text{et}} \tag{23}$$

where $w_{\text{reg}}, w_{\text{cls}}, w_u, w_m, w_e$ balance the contributions of $L_{\text{reg}}, L_{\text{cls}}, L_{u\text{-reg}}, L_{m\text{-reg}}, L_{\text{et}}$, respectively. All weights are equal, and $L_{\text{total}}$ denote the overall loss of training.

# 4 EXPERIMENTS

## 4.1 EXPERIMENTAL SETTINGS

**Datasets & Evaluation Metric** Our method is evaluated on two benchmark datasets such as Argoverse 1 Chang et al. (2019) with 323,557 sequences including 2 seconds of past data and 3 seconds of future prediction, and Argoverse 2 Wilson et al. (2021) with 250,000 scenes providing 5 seconds of observation and 6 seconds of prediction. We assess our approach using the standard metrics $MinADE_k$, $MinFDE_k$, and $MR_k$, with k values of 1 and 6 commonly adopted as benchmarks.

**Implementation Details** Detailed training settings are included in the Appendix A.3 section.

Table 1: Performance under different observation scenarios for Argoverse 2 validation dataset. Best results are shown in **bold**.

| Model | I/N Scenarios | minADE$_1$ | minFDE$_1$ | MR$_1$ | minADE$_6$ | minFDE$_6$ | MR$_6$ |
|---|---|---|---|---|---|---|---|
| DeMo-Orig | Missing + Var. | 2.1672 | 4.9458 | 0.6457 | 0.8889 | 1.5834 | 0.2065 |
| | Var. Obs | 2.0281 | 4.5055 | 0.6329 | 0.8177 | 1.5054 | 0.1971 |
| | Missing Only | 1.6637 | 4.1427 | 0.5944 | 0.6837 | 1.3234 | 0.1666 |
| | Full Obs | 1.6041 | 4.0521 | 0.5889 | 0.6625 | 1.2999 | 0.1623 |
| DeMo-RSD | Missing + Var. | 1.6732 | 4.1777 | 0.5960 | 0.6812 | 1.33277 | 0.1683 |
| | Var. Obs | 1.6613 | 4.1589 | 0.5955 | 0.6769 | 1.3261 | 0.1663 |
| | Missing Only | 1.6747 | 4.1703 | 0.5956 | 0.6863 | 1.3419 | 0.1697 |
| | Full Obs | 1.6341 | 4.1049 | 0.5914 | 0.6719 | 1.3187 | 0.1646 |
| Forecast-mae-Orig | Missing + Var. | 2.1137 | 5.0919 | 0.6735 | 0.8291 | 1.5985 | 0.2130 |
| | Var. Obs | 2.0186 | 4.8589 | 0.6406 | 0.7892 | 1.5182 | 0.2034 |
| | Missing Only | 2.2330 | 5.1860 | 0.6649 | 0.8524 | 1.6087 | 0.2195 |
| | Full Obs | 1.8165 | 4.5536 | 0.6218 | 0.7244 | 1.4273 | 0.1877 |
| Forecast-mae-RSD | Missing + Var. | 1.8246 | 4.5432 | 0.6201 | 0.7335 | 1.4406 | 0.1908 |
| | Var. Obs | 1.8146 | 4.5292 | 0.6204 | 0.7292 | 1.4354 | 0.1902 |
| | Missing Only | 1.8302 | 4.5558 | 0.6224 | 0.7346 | 1.4423 | 0.1918 |
| | Full Obs | 1.8098 | 4.5198 | 0.6200 | 0.7279 | 1.4332 | 0.1905 |
| UIT-Pred (Train-full) | Missing + Var. | 1.5896 | 3.9584 | 0.5778 | 0.6519 | 1.2284 | 0.1580 |
| | Var. Obs | 1.5867 | 3.8597 | 0.5669 | 0.6479 | 1.2331 | 0.1592 |
| | Missing Only | 1.5781 | 3.8146 | 0.5693 | 0.6332 | 1.2377 | 0.1554 |
| | Full Obs | 1.5513 | 3.8643 | 0.5618 | 0.6394 | 1.2483 | 0.1518 |
| UIT-Pred (Train-mixed) | Missing + Var. | 1.5882 | 3.9402 | 0.5717 | 0.6562 | 1.2326 | 0.1551 |
| | Var. Obs | 1.5749 | 3.9281 | 0.5693 | 0.6490 | 1.2316 | 0.1525 |
| | Missing Only | 1.5711 | 3.9059 | 0.5692 | 0.6491 | 1.2481 | 0.1527 |
| | Full Obs | **1.5508** | **3.8784** | **0.5677** | **0.6436** | **1.2455** | **0.1526** |

## 4.2 RESULT AND ANALYSIS

**Performance under Different Observation Settings.** We evaluate the overall performance of the proposed method under various input conditions on the Argoverse 2 dataset, as presented in Table 1. The label *Missing + Var* indicates scenarios where inputs have variable lengths and contain missing data. *Var. Obs* refers to inputs with variable lengths only, without missing data. *Missing Only* denotes inputs that contain missing data. Finally, *Full Obs* represents complete inputs with fixed length. The models *DeMo-Orig* and *Forecast-mae-Orig* are trained exclusively on fully observed, fixed-length inputs and are evaluated across all input conditions to examine their generalization capability. In contrast, *DeMo-RSD* and *Forecast-mae-RSD* are trained on inputs with randomly assigned sequence lengths, where certain time steps are also randomly dropped during training. This training setup is referred to as *RSD* (Random Sequence Drop). *Train-full* denotes training on complete trajectories only, whereas *Train-mixed* denotes training on complete trajectories with simulated intermittent observations (variable-length truncation and random missing values).

The performance of *DeMo-Orig* and *Forecast-mae-Orig* degrades significantly under the *Missing + Var*, *Var. Obs*, and *Missing Only* settings, demonstrating their limitations in handling missing or variable-length inputs. In contrast, *DeMo-RSD* and *Forecast-mae-RSD*, although trained with the *RSD* strategy, do not show notable improvements in these challenging scenarios. This suggests that *RSD* alone is not sufficient to ensure adaptability across diverse input conditions. The proposed model, *UIT-Pred*, demonstrates promising results across all input conditions. (See Appendix A.2.1 for comparisons on the Argoverse 1 dataset.)

Table 2: Comparison of methods on the Argoverse 2 and Argoverse 1 validation sets under variable-length observations.

| Dataset | Method | minADE$_1$ | minFDE$_1$ | MR$_1$ | minADE$_6$ | minFDE$_6$ | MR$_6$ |
|---|---|---|---|---|---|---|---|
| Argoverse 2 | HiVT-Orig | 2.5502 | 6.5586 | 0.7455 | 1.0561 | 2.1093 | 0.3275 |
| | HiVT-RM | 2.2848 | 6.0548 | 0.7249 | 0.9457 | 1.9283 | 0.2994 |
| | HiVT-DTO | 2.2769 | 6.0548 | 0.7275 | 0.9324 | 1.8946 | 0.2903 |
| | HiVT-FLN | 2.2786 | 6.0464 | 0.7240 | 0.9287 | 1.8838 | 0.2891 |
| | HiVT-LaKD | 2.2066 | 5.8769 | 0.7161 | 0.9183 | 1.8686 | 0.2791 |
| | QCNet-Orig | 2.1006 | 5.2219 | 0.6299 | 0.8339 | 1.3849 | 0.1884 |
| | QCNet-RM | 1.7452 | 4.4404 | 0.5957 | 0.7508 | 1.3184 | 0.1671 |
| | QCNet-DTO | 1.7713 | 4.4900 | 0.5979 | 0.7454 | 1.2924 | 0.1671 |
| | QCNet-FLN | 1.6940 | 4.2373 | 0.5808 | 0.7370 | 1.2595 | 0.1596 |
| | QCNet-LaKD | 1.6574 | 4.1505 | 0.5753 | 0.7258 | 1.2420 | 0.1555 |
| | our | **1.5749** | **3.9281** | **0.5693** | **0.6490** | **1.2316** | **0.1525** |
| Argoverse 1 | HiVT-Orig | 1.4733 | 3.1834 | 0.5267 | 0.7255 | 1.0740 | 0.1124 |
| | HiVT-RM | 1.4189 | 3.0599 | 0.5104 | 0.7070 | 1.0447 | 0.1053 |
| | HiVT-DTO | 1.3999 | 3.0262 | 0.5056 | 0.7032 | 1.0350 | 0.1039 |
| | HiVT-FLN | 1.4011 | 3.0288 | 0.5051 | 0.7026 | 1.0325 | 0.1033 |
| | HiVT-LaKD | 1.3317 | 2.8799 | 0.4901 | 0.6807 | 0.9864 | 0.0928 |
| | QCNet-Orig | 1.1656 | 2.4021 | 0.3860 | 0.5791 | 0.7399 | 0.0734 |
| | QCNet-RM | 1.0995 | 2.2550 | 0.3630 | 0.5684 | 0.7115 | 0.0703 |
| | QCNet-DTO | 1.0708 | 2.2303 | 0.3563 | 0.5418 | 0.6848 | 0.0671 |
| | QCNet-FLN | 1.0631 | 2.2083 | 0.3579 | 0.5411 | 0.6680 | 0.0671 |
| | QCNet-LaKD | 0.9982 | 2.0718 | 0.3439 | 0.5240 | 0.6581 | 0.0640 |
| | our | **0.8086** | **1.7343** | **0.2871** | **0.3693** | **0.5901** | **0.0510** |

**Performance under Different Observation Lengths.** The performance of the proposed approach is evaluated in Table 2 with different observation lengths. The *HiVT-Orig* and *QCNet-Orig* models refer to the original versions of HiVT and QCNet trained using fixed-length observed trajectories as input. In contrast, *HiVT-RM* and *QCNet-RM* introduce random masking to the observed trajectories during training to simulate inputs of varying lengths. Variants such as *HiVT-DTO*, *HiVT-FLN*, and *HiVT-LaKD* represent configurations that use HiVT as the backbone, combined with the DTO, FlexiLength Network (FLN), and Length-agnostic Knowledge Distillation (LaKD) modules, respectively. Similarly, *QCNet-DTO*, *QCNet-FLN*, and *QCNet-LaKD* use QCNet as the backbone in combination with the same modules.

The results demonstrate that the proposed model consistently outperforms all baseline methods, including *QCNet-FLN*, *QCNet-LaKD*, *HiVT-FLN*, and *HiVT-LaKD*, across both data sets. As these baselines are specifically designed to handle variable-length inputs, this comparison highlights the generalizability of our approach. Furthermore, our method also surpasses *HiVT-Orig* and *QCNet-Orig*, reinforcing the importance of trajectory prediction frameworks tailored for variable-length observations. Despite the use of random masking in *HiVT-RM* and *QCNet-RM*, our model still achieves superior performance, demonstrating the advantages of our structured design under varying input lengths. Overall, proposed method achieves state-of-the-art results across multiple configurations, confirming its effectiveness and adaptability.

Table 3: Prolonged block occlusions where contiguous segments of $L$ frames are removed from the observation history to mimic long occlusions like parked trucks or tunnels for Argoverse 2 validation dataset.

| $L$ | minADE$_1$ | minFDE$_1$ | MR$_1$ | minADE$_6$ | minFDE$_6$ | MR$_6$ |
|---|---|---|---|---|---|---|
| 10 | 1.6415 | 4.0336 | 0.5831 | 0.6762 | 1.2628 | 0.1594 |
| 20 | 1.6526 | 4.1589 | 0.5944 | 0.6827 | 1.2869 | 0.1563 |
| 30 | 1.6802 | 4.2125 | 0.6026 | 0.7053 | 1.3172 | 0.1669 |
| 40 | 1.7048 | 4.3037 | 0.6183 | 0.7135 | 1.3317 | 0.1725 |

Table 4: Gradient-Based Timestep Removal, rank past timesteps by saliency ($|\partial\mathcal{L}/\partial x_t|$) and remove the top-$k$ most influential frames, yielding the strongest adversarial perturbation for Argoverse 2 validation dataset.

| $k$ | minADE$_1$ | minFDE$_1$ | MR$_1$ | minADE$_6$ | minFDE$_6$ | MR$_6$ |
|---|---|---|---|---|---|---|
| 3 | 1.6470 | 4.0280 | 0.5741 | 0.6748 | 1.2788 | 0.1579 |
| 5 | 1.6669 | 3.9836 | 0.5836 | 0.6898 | 1.2917 | 0.1593 |
| 8 | 1.7110 | 4.1125 | 0.5826 | 0.7053 | 1.3172 | 0.1622 |
| 10 | 1.7521 | 4.2037 | 0.6083 | 0.7343 | 1.3491 | 0.1701 |

**Performance Under Block Occlusion.** To simulate real-world structured missingness, we applied block occlusions in Table 3, removing $L = 10$–$40$ consecutive frames (20–80% of the input).

As $L$ increases, performance degrades marginally because UIT-Pred effectively captures temporal dependencies and underlying motion cues, enabling accurate trajectory prediction across long gaps.

**Performance under Adversarial Missing Patterns.** In Table 4, top-k frames are removed to simulate worst-case missing data. Dropping the top-3 or top-5 frames, already a strong perturbation, has minimal impact on prediction, indicating that UIT-Pred does not rely on a small set of critical frames. Performance degrades only under extreme removals (k=8–10), where large high-gradient regions are lost, yet the drop remains gradual, showing strong resilience even under adversarial missing patterns.

**Ablation of Each Component.** Table 5 presents a component study of the proposed model, evaluating the contributions of the Time-Aware Input Representation (TAIR), BTD-Mamba (BTD-M.), and the Predictor Module (PM). When BTD-Mamba is not used, the model depened only on the forward Mamba module without Time Decay (TD) and when the PM

Table 5: Component Study of Proposed Model for Argoverse 2 validation dataset

| ID | TAIR | BTD-M. | PM | minADE$_1$ | minFDE$_1$ | MR$_1$ | minADE$_6$ | minFDE$_6$ | MR$_6$ |
|---|---|---|---|---|---|---|---|---|---|
| 1 | | | | 1.8528 | 4.6447 | 0.6442 | 0.8189 | 1.4668 | 0.1915 |
| 2 | √ | | | 1.8071 | 4.3175 | 0.6257 | 0.7538 | 1.4381 | 0.1830 |
| 3 | | √ | | 1.7847 | 4.2176 | 0.6137 | 0.7381 | 1.4169 | 0.1798 |
| 4 | | | √ | 1.7516 | 4.0163 | 0.6149 | 0.7257 | 1.4037 | 0.1705 |
| 5 | √ | √ | | 1.6928 | 3.9826 | 0.6037 | 0.7081 | 1.3569 | 0.1629 |
| 6 | √ | | √ | 1.6683 | 3.9471 | 0.5973 | 0.6822 | 1.3244 | 0.1648 |
| 7 | | √ | √ | 1.6528 | 3.9714 | 0.5901 | 0.6962 | 1.3062 | 0.1631 |
| 8 | √ | √ | √ | 1.5882 | 3.9402 | 0.5717 | 0.6562 | 1.2326 | 0.1551 |

is excluded, separate MLPs are used for each prediction. The model performs the worst when none of the components are used (ID-1), highlighting their necessity. Introducing at least one component (ID-2, 3, 4) leads to noticeable performance gains, as each individual module provides valuable information. When any two components are combined (ID-5, 6, 7), the model benefits from their complementary strengths, further improving performance. Finally, the best results are achieved when all three components are used together (ID-8), highlighting their benefit of integration.

**Alternative Decay Parameterizations.** We evaluate the model under four alternative temporal-decay parameterizations, exponential decay $[f_f, f_b]_{exp} = \frac{1}{\exp(\phi_s(\mathbf{dt}^{(f,b)}; W_s))}$, $[f_f, f_b]_\sigma = \sigma(\phi_s(dt^{f,b}; W_d))$, sigmoid gating $[f_f, f_b]_\sigma = \sigma(\phi_s(dt^{f,b}; W_d))$, linear-clipped decay $[f_f, f_b]_{lin} = \text{clip}(1 - \beta \cdot \phi_s(dt^{(f,b)}; W_d), 0, 1)$, and softplus-inverse decay $[f_f, f_b]_{sp} = \frac{1}{1 + \text{softplus}(\phi_s(dt^{(f,b)}; W_d))}$, as summarized in Table 6. Here, $f_f$ and $f_b$ denote the forward and backward temporal scaling factors, respectively, applied to the features. The function $\phi_s(\cdot)$ is a projection function parameterized by weights $W_d$, implemented as a multi-layer perceptron (MLP). The operator

Table 6: Performance under alternative temporal-decay parameterizations across different observation scenarios on the Argoverse 2 validation dataset.

| Model | I/N Scenarios | minADE$_1$ | minFDE$_1$ | MR$_1$ | minADE$_6$ | minFDE$_6$ | MR$_6$ |
|---|---|---|---|---|---|---|---|
| Sigmoid Gating $[f_f, f_b]_\sigma$ | Missing + Var. | 1.6087 | 3.9887 | 0.5749 | 0.6637 | 1.2851 | 0.1562 |
| | Var. Obs | 1.5845 | 3.9407 | 0.5683 | 0.6568 | 1.2719 | 0.1523 |
| | Missing Only | 1.5793 | 3.9339 | 0.5702 | 0.6591 | 1.2705 | 0.1515 |
| | Full Obs | 1.5659 | 3.8939 | 0.5669 | 0.6575 | 1.2680 | 0.1518 |
| Linear Clipped $[f_f, f_b]_{lin}$ | Missing + Var. | 1.6139 | 4.0112 | 0.5769 | 0.6595 | 1.2740 | 0.1552 |
| | Var. Obs | 1.5923 | 3.9789 | 0.5772 | 0.6535 | 1.2636 | 0.1536 |
| | Missing Only | 1.5788 | 3.9346 | 0.5727 | 0.6517 | 1.2580 | 0.1528 |
| | Full Obs | 1.5682 | 3.9264 | 0.5737 | 0.6484 | 1.2540 | 0.1512 |
| Softplus Inverse $[f_f, f_b]_{sp}$ | Missing + Var. | 1.6117 | 4.0067 | 0.5731 | 0.6609 | 1.2798 | 0.1543 |
| | Var. Obs | 1.5814 | 3.9506 | 0.5685 | 0.6520 | 1.2669 | 0.1528 |
| | Missing Only | 1.5692 | 3.9120 | 0.5677 | 0.6555 | 1.2628 | 0.1528 |
| | Full Obs | 1.5576 | 3.8978 | 0.5647 | 0.6472 | 1.2566 | 0.1509 |
| Exponential Decay $[f_f, f_b]_{exp}$ | Missing + Var. | 1.5882 | 3.9402 | 0.5717 | 0.6562 | 1.2326 | 0.1551 |
| | Var. Obs | 1.5749 | 3.9281 | 0.5693 | 0.6490 | 1.2316 | 0.1525 |
| | Missing Only | 1.5711 | 3.9059 | 0.5692 | 0.6491 | 1.2481 | 0.1527 |
| | Full Obs | **1.5508** | **3.8784** | **0.5677** | **0.6436** | **1.2455** | **0.1526** |

$\sigma(\cdot)$ denotes the sigmoid function and $\beta$ is a constant set to 1. We observe that exponential decay performs best across all I/N scenarios because its continuous-time form, with range $(0, \infty)$, provides smooth, non-saturating attenuation that preserves temporal cues across all gap sizes and remains stable for $dt$. In contrast, the sigmoid output is bounded in $(0, 1)$, so moderate or large $dt$ values push it rapidly toward saturation, compressing mid-range differences and reducing sensitivity. Linear-clipped decay, restricted to $[0, 1]$, suppresses information too abruptly; once the linear term exceeds this interval, clipping prevents representing stronger decay, producing hard cutoffs even for moderately large $dt$. Softplus-inverse decay, also in the $(0, 1)$ range, behaves similarly to exponential under full observations but decays more aggressively for large $dt$. This strong

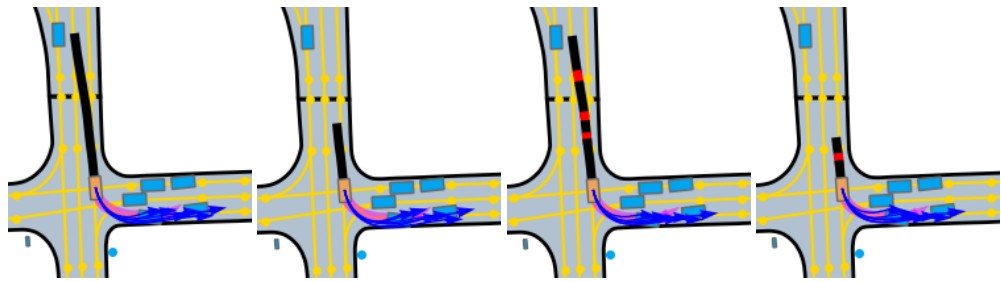

Figure 3: Qualitative results of the proposed model with varying input observations. Black: observed history; red: missing points; magenta: ground truth future; blue: predicted future trajectory.

suppression helps in missing-only settings by down-weighting outdated information, but leads to slightly worse performance in mixed scenarios where such aggressive decay removes useful context.

**Decay component of BTD-Mamba.**
We assess the learned decay in BTD-Mamba using three variants: (i) a fixed exponential decay (constant 0.5), (ii) no decay, and (iii) the learned decay. As shown in Table 7, both fixed and no-decay settings yield higher minADE, minFDE, and MR under intermittent observations, while the learned decay maintains the

Table 7: Impact of fixed, no, and learned decay in BTD-Mamba for the Argoverse 2 validation set.

| Method | $minADE_1$ | $minFDE_1$ | $MR_1$ | $minADE_6$ | $minFDE_6$ | $MR_6$ |
|---|---|---|---|---|---|---|
| Fixed-decay | 1.6824 | 4.1138 | 0.6036 | 0.7045 | 1.3459 | 0.1858 |
| No-decay | 1.6529 | 4.0284 | 0.5903 | 0.6925 | 1.3163 | 0.1628 |
| Learned-decay | 1.5882 | 3.9402 | 0.5717 | 0.6562 | 1.2326 | 0.1551 |

best performance. Fixed decay fails because a single rate cannot capture diverse motion patterns or gap lengths, and no decay performs worst due to stale states persisting over long gaps. These results show that adaptive, learned decay is essential for robustness under intermittent observations.

**Computational Efficiency.** UIT-Pred shows strong computational efficiency in Table 8 across both Argoverse datasets. With 7.5M parameters, training requires 900–1200 min on two RTX A5000 GPUs, while inference uses 6.5M parameters and achieves real-time speeds of 2.1–2.4 ms per sample at 0.51–0.52 GFLOPs, despite the bidirectional SSM and transformer architecture.

Table 8: Computational efficiency of the proposed model using two NVIDIA RTX A5000 GPUs. Abbr.: TTT - Total Training Time, IT/S - Inference Time per Sample, TP - Training Parameters, IP - Inference Parameters, BS - Batch Size, F/S - FLOPs per Sample.

| Datasets | TTT (min) | IT/S (ms) | TP (M) | IP (M) | BS | F/S (G) |
|---|---|---|---|---|---|---|
| Argoverse 2 | 900 | 2.11 | 7.53 | 6.55 | 128 | 0.52 |
| Argoverse 1 | 1200 | 2.45 | 7.48 | 6.50 | 128 | 0.51 |

**Qualitative Result.** Figure 3 shows the proposed model's performance during a left-turn maneuver at an intersection in the Argoverse 2 dataset. The model effectively handles all input conditions, aligning well with the real-world requirements of autonomous driving. Additional qualitative results across diverse driving scenarios are provided in Appendix A.3.

## 5 CONCLUSION

In this work, we introduce UIT-Pred, a universal architecture that addresses real-world challenges in autonomous driving by handling diverse input types for trajectory prediction. We propose a time-aware input representation that helps the model focus on motion dynamics across diverse input conditions rather than absolute durations. Furthermore, we extend a state space model to develop the BTD-Mamba module and introduce a novel predictor, jointly capturing complex temporal dynamics to enhance trajectory prediction accuracy. Comprehensive experiments on the Argoverse 1 and Argoverse 2 datasets demonstrate effectiveness of our approach.

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

# A APPENDIX

## A.1 EXPERIMENTAL SETTINGS

**Input Representation for Neighbours and Lane information.** Using the vectorized representation approach Zhang et al. (2024), the trajectories of all agents and the geometric representation of lane segments are modeled as polylines composed of interconnected points. We employ an agent-centric normalization strategy Cheng et al. (2023), which transforms all inputs into a coordinate system centered on the target agent. The historical trajectories of $N_a$ agents are represented as $\mathbf{X}^n$, which include coordinates and velocity changes over a length of $T_{\max}$ timesteps. Furthermore, we incorporate two time-related features, scaled timestamps to manage varying time ranges, and inter-observation interval features to capture timing differences between observations, to construct the time-aware input representation $h^n$.

Lane segments are encoded as $\mathbf{h}^l$, which includes the number of lane segments within a specified radius around the target agent, the number of points in each polyline, and lane features such as coordinates and availability. All coordinates within each lane segment are normalized relative to their geometric centers, providing a standardized reference frame for subsequent processing and analysis Bharilya et al. (2025).

**Evaluation Metric.** We assess our approach using several widely accepted metrics in trajectory prediction research Wilson et al. (2021). The $MinADE_k$ metric calculates the average Euclidean distance between the predicted trajectories and the actual ground truth paths. The $MinFDE_k$ metric, on the other hand, measures the prediction error specifically at the endpoints of the trajectories. To evaluate failure rates, the miss rate ($MR_k$) counts instances where the endpoint error $MinFDE_k$ exceeds a threshold of 2 meters. In these metrics, $k$ denotes the number of trajectory modes predicted, with evaluations conducted for both single-mode predictions ($k = 1$) and multi-modal predictions ($k = 6$).

**Implementation Details.** The framework is implemented in PyTorch and trained on an NVIDIA RTX A5000 GPU. Models are trained end-to-end for 60 epochs using the AdamW optimizer, with a batch size of 128, a learning rate of 0.001, and a weight decay of 0.01. We use a cosine learning rate schedule with a 10-epoch warm-up phase. An agent-centric coordinate system samples scene elements within a 150-meter radius around the agents of interest. The embedding dimension $d$ is set to 128. Each Mamba block contains 4 layers, the Transformer encoder has 5 layers, and the Transformer decoder $\mathcal{D}(\cdot, \cdot, \cdot)$ in the prediction module is used for cross-attention, where the first argument is the query, the second is the key, and the third is the value. It consists of two layers.

**Formulation of Loss Functions.** To optimize the model, we use the Huber loss for trajectory regression $L_{reg}$ and cross-entropy loss for confidence classification $L_{\text{cls}}$. A winner-take-all strategy is applied, optimizing only the best prediction while minimizing the average error relative to the ground truth,

$$L_{reg} = \min_{k \in \{1,2,...,6\}} \left( \sum_{t=1}^{t_{pred}} \sum_{c=1}^{2} L_{\text{hl}}(Y_{gt}^{(t,c)}, \hat{Y}_k^{(t,c)}) \right) \tag{24}$$

where $\hat{Y}_k^{(t,c)}$ denotes the predicted future trajectory for mode $k$ at timestamp $t$ along coordinate $c$, $Y_{gt}^{(t,c)}$ represents the corresponding ground truth, $t_{\text{pred}}$ is the total number of future time steps, and $k$ indicates the number of predicted modes. For confidence classification, we apply the cross-entropy loss,

$$L_{\text{cls}} = \sum_{k=1}^{K} \left( \mathbb{I}[Y_{gt}] \log(\pi_k) + (1 - \mathbb{I}[Y_{gt}]) \log(1 - \hat{\pi}_k) \right) \tag{25}$$

where $\hat{\pi}_k$ is the predicted probability for the $k$-th trajectory, and $\mathbb{I}[Y_{gt}]$ is an indicator function that equals 1 if the $k$-th trajectory is closest to the ground truth, and 0 otherwise.

To supervise the missing-step temporal embedding, the loss $L_{\text{m-rg}}$ is computed as,

$$L_{\text{m-rg}} = \sum_{m=1}^{t_m} \sum_{c=1}^{2} L_{\text{hl}} \left( x_{\text{gt}}^{(m,c)}, \hat{x}_{\text{m}}^{(m,c)} \right) \tag{26}$$

where $t_m$ is the number of missing data in the observation. Here, $m$ indexes the missing points, and $x_{\text{gt}}^{(m,c)}$ and $\hat{x}_{\text{m}}^{(m,c)}$ denote the ground truth and predicted values of the missing data, respectively, along dimension $c$. The function $L_{\text{hl}}$ refers to the Huber loss used for regression.

Moreover, to supervise the variable-step temporal embedding, the loss $L_{\text{v-rg}}$ is defined as,

$$L_{\text{v-rg}} = \sum_{v=1}^{t_{uv}} \sum_{c=1}^{2} L_{\text{hl}} \left( x_{\text{gt}}^{(v,c)}, \hat{x}_{\text{v}}^{(v,c)} \right), \tag{27}$$

where $t_{uv}$ denotes the number of unobserved time steps with variable length. Here, $v$ indexes the un-observed tokens, and $x_{\text{gt}}^{(v,c)}$ and $\hat{x}_{\text{v}}^{(v,c)}$ represent the ground truth and predicted values, respectively, along dimension $c$.

**Endpoint loss.** To predict endpoints, we utilize a dynamic multi-layer perceptron (MLP) with weights that are adaptively generated based on the input, referred to as the *adaptive MLP*. The adaptive MLP takes as input the agent features $[E^t, F^n]$ and meta-information $mi$ of all agents. The meta-information includes the agent's position and normalized velocity at last observed times-tamps. These inputs are concatenated and passed through an MLP with learnable parameters $W_{\text{feat},1}, W_{\text{feat},2}$ and biases $b_{\text{feat},1}, b_{\text{feat},2}$, to obtain a latent representation, $\tilde{f}$,

$$\tilde{f} = \varphi \left( W_{\text{feat},2}\, \varphi \left( W_{\text{feat},1}[E^t, F^n]; mi] + b_{\text{feat},1} \right) + b_{\text{feat},2} \right) \tag{28}$$

with $\varphi$ denoting the ReLU activation. Subsequently, two sets of dynamic weights $W_1$ and $W_2$ are generated by applying learnable linear transformations $W_{d1}$ and $W_{d2}$ to $\tilde{f}$, reshaped accordingly.

$$W_1 = \text{reshape}\left(W_{d1} \cdot \tilde{f} + b_{d1}\right) \tag{29}$$

$$W_2 = \text{reshape}\left(W_{d2} \cdot \tilde{f} + b_{d2}\right) \tag{30}$$

The first hidden layer activations $F_{d1}$ are computed by applying a linear transformation $W_1$ to the input feature $f$, followed by layer normalization and a non-linear activation function $\varphi$. The final prediction $\hat{y}_{\text{ep}}$ is then obtained by applying a second linear transformation $W_2$ to $F_{d1}$,

$$F_{d1} = \varphi\left(\text{LayerNorm}(W_1 \cdot f)\right), \tag{31}$$

$$\hat{y}_{\text{ep}} = W_2 \cdot F_{d1} \tag{32}$$

This formulation enables dynamic adaptation of the prediction weights conditioned on the input features and meta information, allowing the model to flexibly predict agent endpoints. To improve the accuracy of endpoint predictions, we employ a dedicated loss defined as,

$$L_{\text{et}} = \sum_{n=1}^{N} \sum_{c=1}^{2} L_{\text{hl}} \left( Y_{\text{gt},n}^{(t_{\text{pred}},c)}, \hat{ep}_n^{(t_{\text{pred}},c)} \right) \tag{33}$$

where $L_{\text{et}}$ measures the discrepancy between the predicted endpoint $\hat{ep}_n$ and the ground truth end-point $Y_{\text{gt},n}^{(t_{\text{pred}},c)}$ of the $n^{th}$ agent, computed using the Huber loss function.

## A.2   MORE EXPERIMENTAL RESULTS

### A.2.1   PERFORMANCE UNDER DIFFERENT OBSERVATION SETTINGS.

The performance of the proposed model on the Argoverse 1 dataset under different input conditions is shown in Table 9.

Table 9: Performance under different observation scenarios for Argoverse 1 validation dataset

| Model | I/N Scenarios | $minADE_1$ | $minFDE_1$ | $MR_1$ | $minADE_6$ | $minFDE_6$ | $MR_6$ |
|---|---|---|---|---|---|---|---|
| DeMo-Orig | Missing + Var. | 2.3924 | 4.6935 | 0.6172 | 0.9052 | 1.4566 | 0.1751 |
| | Var. Obs | 2.1506 | 4.2475 | 0.5993 | 0.8311 | 1.3058 | 0.5993 |
| | Missing Only | 3.4942 | 6.6470 | 0.7808 | 1.4078 | 2.2559 | 0.3238 |
| | Full Obs | 1.2903 | 2.7863 | 0.46495 | 0.5926 | 0.9534 | 0.0830 |
| DeMo-RSD | Missing + Var. | 1.5100 | 3.1800 | 0.5174 | 0.6504 | 1.0569 | 0.1010 |
| | Var. Obs | 1.5798 | 3.3039 | 0.5380 | 0.6538 | 1.0576 | 0.1022 |
| | Missing Only | 2.7708 | 1.8213 | 0.7421 | 1.0927 | 1.8213 | 0.2572 |
| | Full Obs | 1.3999 | 2.9870 | 0.5030 | 0.6145 | 0.9946 | 0.0918 |
| Forecast-mae-Orig | Missing + Var. | 1.7647 | 3.6791 | 0.6153 | 0.7315 | 1.2033 | 0.1175 |
| | Var. Obs | 1.4734 | 3.1700 | 0.5340 | 0.6669 | 1.0936 | 0.0964 |
| | Missing Only | 2.2504 | 4.4974 | 0.6836 | 0.8654 | 1.4303 | 0.1757 |
| | Full Obs | 1.3470 | 2.9207 | 0.0901 | 0.6223 | 1.0222 | 0.0901 |
| Forecast-mae-RSD | Missing + Var. | 1.4679 | 3.1545 | 0.5226 | 0.6643 | 1.0849 | 0.0952 |
| | Var. Obs | 1.4576 | 3.1298 | 0.5243 | 0.6599 | 1.0759 | 0.0945 |
| | Missing Only | 1.7459 | 3.6371 | 0.6066 | 0.7206 | 1.1881 | 0.1172 |
| | Full Obs | 1.4338 | 3.0894 | 0.5120 | 0.6554 | 1.0707 | 0.0949 |
| Our | Missing + Var. | 0.8212 | 1.7826 | 0.3028 | 0.3956 | 0.6018 | 0.0698 |
| | Var. Obs | 0.8086 | 1.7343 | 0.2871 | 0.3693 | 0.5901 | 0.0510 |
| | Missing Only | 0.8123 | 1.7365 | 0.2816 | 0.3756 | 0.5902 | 0.0511 |
| | Full Obs | 0.7943 | 1.7029 | 0.2718 | 0.3346 | 0.5726 | 0.0467 |

### A.2.2 PERFORMANCE UNDER BURST DROPOUT.

We introduce burst-interval missingness (Table 10), where multiple short gaps (5–30 frames) are distributed throughout the sequence to mimic intermittent sensor dropout or brief occlusions. In the burst-size column, *max* indicates the maximum possible length of each dropped segment, while *fix* specifies a constant number of frames per burst. The *max* setting produces higher errors because it introduces more severe and occasionally longer gaps, making temporal continuity harder to reconstruct. UIT-Pred remains stable even under this more challenging regime, whereas the *fix* setting is easier due to its consistent gap lengths, which lead to more predictable agent trajectories. Notably, UIT-Pred can be further improved by incorporating burst-shape augmentation without requiring any architectural modifications.

### A.2.3 PERFORMANCE UNDER INTERSECTION-BASED DROPPING.

To evaluate the model under structured occlusions, Table 11 reports results for two forms of intersection-based dropping. (1) Probability-based dropping removes each point with high probability (0.8) when it lies inside an intersection polygon and with low probability (0.1) otherwise. (2)Intersection block occlusion removes a continuous block of (L) points once the trajectory enters an intersection. Despite differing in how drops are applied, both methods produce nearly identical errors because they focus missing segments around intersections, regions with inherently complex motion, leading to temporal gaps of similar effective severity. UIT-Pred remains robust under both settings, demonstrating strong generalization across localized temporal dropout.

### A.2.4 ABLATION STUDY OF BTD-MAMBA COMPONENTS.

Table 12 presents an ablation study analyzing different configurations of the BTD-Mamba module. Using only the forward Mamba (Fwd) or backward Mamba (Bwd) results in similar performance, with slightly better results for Fwd. Combining both directions (Fwd+Bwd) improves performance across all metrics, indicating that bidirectional context benefits trajectory modeling. The addition of Time Decay (TD) further enhances performance when combined with either Fwd or Bwd, showing that temporal relationships contribute useful dynamics. The best performance is achieved when

Table 10: Burst-interval missingness where repeated short gaps in frames within the input sequences, simulating brief occlusions in the Argoverse 2 validation dataset.

| $\#Frames$ | Burst Size | $minADE_1$ | $minFDE_1$ | $MR_1$ | $minADE_6$ | $minFDE_6$ | $MR_6$ |
|---|---|---|---|---|---|---|---|
| 5(10%) | max =2 | 1.5724 | 3.9096 | 0.5655 | 0.6490 | 1.2502 | 0.1554 |
| | fix =2 | 1.5662 | 3.8932 | 0.5659 | 0.6480 | 1.2498 | 0.1528 |
| 10(20%) | max =5 | 1.6176 | 3.9759 | 0.5739 | 0.6678 | 1.2717 | 0.1555 |
| | fix =5 | 1.5909 | 3.9390 | 0.5696 | 0.6570 | 1.2631 | 0.1554 |
| 20(40%) | max =10 | 1.7162 | 4.1129 | 0.5832 | 0.7071 | 1.3108 | 0.1620 |
| | fix =10 | 1.6933 | 4.0904 | 0.5812 | 0.7019 | 1.3089 | 0.1629 |
| 30(60%) | max =10 | 2.0012 | 4.4932 | 0.6111 | 0.8406 | 1.4373 | 0.1830 |
| | fix =10 | 1.9376 | 4.4227 | 0.6051 | 0.8092 | 1.4159 | 0.1802 |

Table 11: Comparison of two intersection-based occlusion mechanisms, probability-based dropping and block occlusion, on the Argoverse 2 validation dataset.

| Method | $minADE_1$ | $minFDE_1$ | $MR_1$ | $minADE_6$ | $minFDE_6$ | $MR_6$ |
|---|---|---|---|---|---|---|
| Probability-based Dropping | 1.6552 | 4.0185 | 0.5926 | 0.6659 | 1.2894 | 0.1593 |
| Block occlusion | 1.6431 | 3.9926 | 0.5812 | 0.6623 | 1.2773 | 0.1576 |

all three components such as Fwd, Bwd, and TD are integrated, forming the complete BTD-Mamba module. This full configuration achieves the lowest minADE and minFDE, as well as the lowest miss rate, demonstrating the complementary nature of bidirectional processing and temporal differencing.

A.2.5 ABLATION STUDY ON PREDICTOR MODULE COMPONENTS.

Table 13 evaluates the impact of three components in the predictor module: State Encoding for Intermittent Trajectories (SEIT), Multi-Modal Trajectory Distributions (MMTD), and Unified Mode Fusion (UMF). When MMTD is not used, separate MLPs replace it for mode and state prediction. The baseline model without these components (ID-1) performs the worst. Incorporating each component individually (ID-2 to ID-4) yields moderate improvements, indicating their standalone effectiveness. Combinations of two components (ID-5 to ID-7) further enhance performance, demonstrating complementary strengths. The full model with all three components enabled (ID-8) achieves the best results, confirming that TEIT, MMTD, and UMF together significantly improve trajectory prediction accuracy across all metrics.

A.2.6 ABLATION STUDY ON THE DEPTH OF BTD-MAMBA AND TRANSFORMER ENCODER.

Table 14 investigates the impact of varying the depth i.e., the number of stacked layers, of the BTD-Mamba module and the Transformer Encoder on model performance. Increasing the depth of both modules generally improves results, as seen when moving from 3 to 4 layers, leading to reduced minADE, minFDE, and Miss Rate (MR). The best performance is observed with 4 layers of BTD-Mamba and 5 layers of the Transformer Encoder, achieving the lowest across all metrics. Moreover, performance slightly declines when the BTD-Mamba depth is increased to 5 layers alongside 5

Table 12: Component Study of BTD-Mamba for Argoverse 2 validation dataset.

| Method | $minADE_1$ | $minFDE_1$ | $MR_1$ | $minADE_6$ | $minFDE_6$ | $MR_6$ |
|---|---|---|---|---|---|---|
| Fwd | 1.6816 | 4.1148 | 0.6027 | 0.7011 | 1.3463 | 0.1804 |
| Bwd | 1.6901 | 4.1757 | 0.6134 | 0.7128 | 1.3524 | 0.1749 |
| Fwd+Bwd | 1.6529 | 4.0284 | 0.5903 | 0.6925 | 1.3163 | 0.1628 |
| Fwd+TD | 1.6425 | 4.0143 | 0.5901 | 0.6911 | 1.3047 | 0.1609 |
| Bwd+TD | 1.6546 | 4.0112 | 0.5928 | 0.6836 | 1.3142 | 0.1628 |
| BTD-Mamba | 1.5882 | 3.9402 | 0.5717 | 0.6562 | 1.2326 | 0.1551 |

Table 13: Component Study of Predictor Module for Argoverse 2 validation dataset.

| ID | SEIT | MMTD | UMF | $minADE_1$ | $minFDE_1$ | $MR_1$ | $minADE_6$ | $minFDE_6$ | $MR_6$ |
|---|---|---|---|---|---|---|---|---|---|
| 1 | - | - | - | 1.6474 | 4.2143 | 0.6245 | 0.6957 | 1.2764 | 0.1802 |
| 2 | $\checkmark$ | - | - | 1.6349 | 4.2094 | 0.6137 | 0.6835 | 1.2638 | 0.1782 |
| 3 | - | $\checkmark$ | - | 1.6298 | 4.2072 | 0.6048 | 0.6804 | 1.2477 | 0.1756 |
| 4 | - | - | $\checkmark$ | 1.6164 | 4.1537 | 0.6013 | 0.6787 | 1.2416 | 0.1726 |
| 5 | $\checkmark$ | $\checkmark$ | - | 1.6064 | 4.1121 | 0.5912 | 0.6765 | 1.2569 | 0.1683 |
| 6 | $\checkmark$ | - | $\checkmark$ | 1.6092 | 4.0154 | 0.5936 | 0.6627 | 1.2535 | 0.1647 |
| 7 | - | $\checkmark$ | $\checkmark$ | 1.5914 | 3.9668 | 0.5805 | 0.6613 | 1.2476 | 0.1589 |
| 8 | $\checkmark$ | $\checkmark$ | $\checkmark$ | 1.5882 | 3.9402 | 0.5717 | 0.6562 | 1.2326 | 0.1551 |

Table 14: Depth Study of BTD-Mamba and Transformer Encoder for Argoverse 2 validation dataset.

| BTD-M | T-Enc | $minADE_1$ | $minFDE_1$ | $MR_1$ | $minADE_6$ | $minFDE_6$ | $MR_6$ |
|---|---|---|---|---|---|---|---|
| 3 | 3 | 1.6226 | 4.0151 | 0.5926 | 0.6844 | 1.2531 | 0.1629 |
| 4 | 4 | 1.6048 | 3.9197 | 0.5802 | 0.6614 | 1.2494 | 0.1600 |
| 4 | 5 | 1.5882 | 3.9402 | 0.5717 | 0.6562 | 1.2326 | 0.1551 |
| 5 | 5 | 1.5901 | 3.9937 | 0.5826 | 0.6632 | 1.2471 | 0.1622 |

Transformer layers, suggesting a trade-off where excessive depth in BTD-Mamba fails to yield further benefits. Overall, a moderate depth configuration balances model complexity and predictive accuracy effectively.

### A.2.7 ABLATION STUDY ON THE IMPACT OF AUXILIARY LOSSES.

Table 15 presents an ablation study that evaluate contribution of three auxiliary losses: the endpoint loss ($L_{et}$), regression loss over variable-step temporal embedding ($L_{u\text{-}rg}$), and regression loss for missing-step temporal embedding ($L_{m\text{-}rg}$). The baseline model without any auxiliary loss (ID-1) shows the weakest performance across all metrics. Introducing each loss individually (ID-2 to ID-4) yields consistent improvements, demonstrating their individual effectiveness. Both $L_{u\text{-}rg}$ and $L_{m\text{-}rg}$ lead to greater gains than $L_{et}$. Combining two of the losses (ID-5 to ID-7) further improves performance, showing their complementary effects. The best results are obtained when all three auxiliary losses are applied simultaneously (ID-8), achieving the lowest minADE, minFDE, and MR. These findings confirm that auxiliary supervision strengthens the model's ability to learn more accurate trajectory representations.

### A.2.8 ABLATION STUDY ON THE SENSITIVITY OF LOSS WEIGHTS.

We conducted a loss-weight sensitivity analysis, as shown in Table 16, in which we (i) report performance using fixed weighting schedules, shown in the first and last rows with varying weights for each loss and (ii) evaluate a learned, uncertainty-based weighting scheme in the second row. In the uncertainty-based weighting scheme, we introduce learnable uncertainty parameters $s_i$ for each

Table 15: Impact of Auxilary Losses for Argoverse 2 validation dataset.

| ID | $L_{et}$ | $L_{u\text{-}rg}$ | $L_{m\text{-}rg}$ | $minADE_1$ | $minFDE_1$ | $MR_1$ | $minADE_6$ | $minFDE_6$ | $MR_6$ |
|---|---|---|---|---|---|---|---|---|---|
| 1 | - | - | - | 1.6284 | 4.2591 | 0.6137 | 0.6814 | 1.2641 | 0.1795 |
| 2 | $\checkmark$ | - | - | 1.6211 | 4.1918 | 0.6093 | 0.6787 | 1.2601 | 0.1725 |
| 3 | - | $\checkmark$ | - | 1.6159 | 4.0166 | 0.5935 | 0.6749 | 1.2538 | 0.1617 |
| 4 | - | - | $\checkmark$ | 1.6117 | 4.0137 | 0.5931 | 0.6732 | 1.2546 | 0.1629 |
| 5 | $\checkmark$ | $\checkmark$ | - | 1.6026 | 4.0118 | 0.5874 | 0.6726 | 1.2525 | 0.1658 |
| 6 | $\checkmark$ | - | $\checkmark$ | 1.5984 | 3.9971 | 0.5846 | 0.6815 | 1.2437 | 0.1604 |
| 7 | - | $\checkmark$ | $\checkmark$ | 1.5907 | 3.9473 | 0.5824 | 0.6810 | 1.2429 | 0.1598 |
| 8 | $\checkmark$ | $\checkmark$ | $\checkmark$ | 1.5882 | 3.9402 | 0.5717 | 0.6562 | 1.2326 | 0.1551 |

Table 16: Weight sensitivity study of loss for Argoverse 2 validation dataset.

| $w_{\text{reg}}$ | $w_{\text{cls}}$ | $w_u$ | $w_m$ | $w_e$ | minADE$_1$ | minFDE$_1$ | MR$_1$ | minADE$_6$ | minFDE$_6$ | MR$_6$ |
|---|---|---|---|---|---|---|---|---|---|---|
| 1 | 1 | 0.5 | 0.5 | 0.2 | 1.6204 | 4.0284 | 0.5723 | 0.6569 | 1.2664 | 0.1554 |
| - | - | - | - | - | 1.6214 | 4.1926 | 0.6098 | 0.6783 | 1.2616 | 0.1722 |
| 1 | 1 | 1 | 1 | 1 | 1.5882 | 3.9402 | 0.5717 | 0.6562 | 1.2326 | 0.1551 |

task-specific loss, regression ($L_{\text{reg}}$), classification ($L_{\text{cls}}$), variable-step temporal embedding ($L_v$), missing-step temporal embedding ($L_m$), and the endpoint loss ($L_e$), to automatically balance multi-task training. Each parameter is initialized to 0 (corresponding to unit variance) and optimized jointly with the model. Under this scheme, the combined loss is defined as follows,

$$\mathcal{L}_{\text{total}} = \sum_i \frac{1}{2} \left( L_i \cdot \exp(-s_i) + s_i \right) \tag{34}$$

where $i \in \{\text{reg}, \text{cls}, v, m, e\}$. This approach learns the relative weighting of each task based on observed uncertainty, removing the need for manual tuning. The inferred standard deviations, $\sigma_i = \sqrt{\exp(s_i)}$, provide interpretable uncertainty measures.

Notably, the uniform weighting configuration ($w_{\text{reg}} = w_{\text{cls}} = w_u = w_m = w_e = 1$) achieves the best overall performance, with the lowest minADE, minFDE, and MR across both top-1 and top-6 predictions. This suggests that while uncertainty-based weighting can adaptively balance tasks, equal weighting is sufficient in our setup, providing strong performance without additional complexity. The results highlight the robustness of our multi-task training formulation and indicate that all task-specific losses contribute meaningfully to trajectory prediction.

## A.3 MORE QUALITATIVE RESULTS

The qualitative results are presented in Figure 4, showcasing the performance of the proposed model across diverse driving scenarios. The first row illustrates an intersection scenario where the agent executes a right turn, requiring awareness of both lane geometry and surrounding context. The second row demonstrates the agent's behavior on a curved path, highlighting the model's ability to capture smooth trajectory changes over time. The third row presents a straight-driving scenario in dense traffic, where the model must accurately predict future motion despite limited maneuvering space and potential occlusions. Across all scenarios, the columns depict different input conditions, including variable-length observations and missing data. The proposed model consistently produces coherent and accurate trajectory predictions, demonstrating its adaptability to a wide range of real-world input conditions.

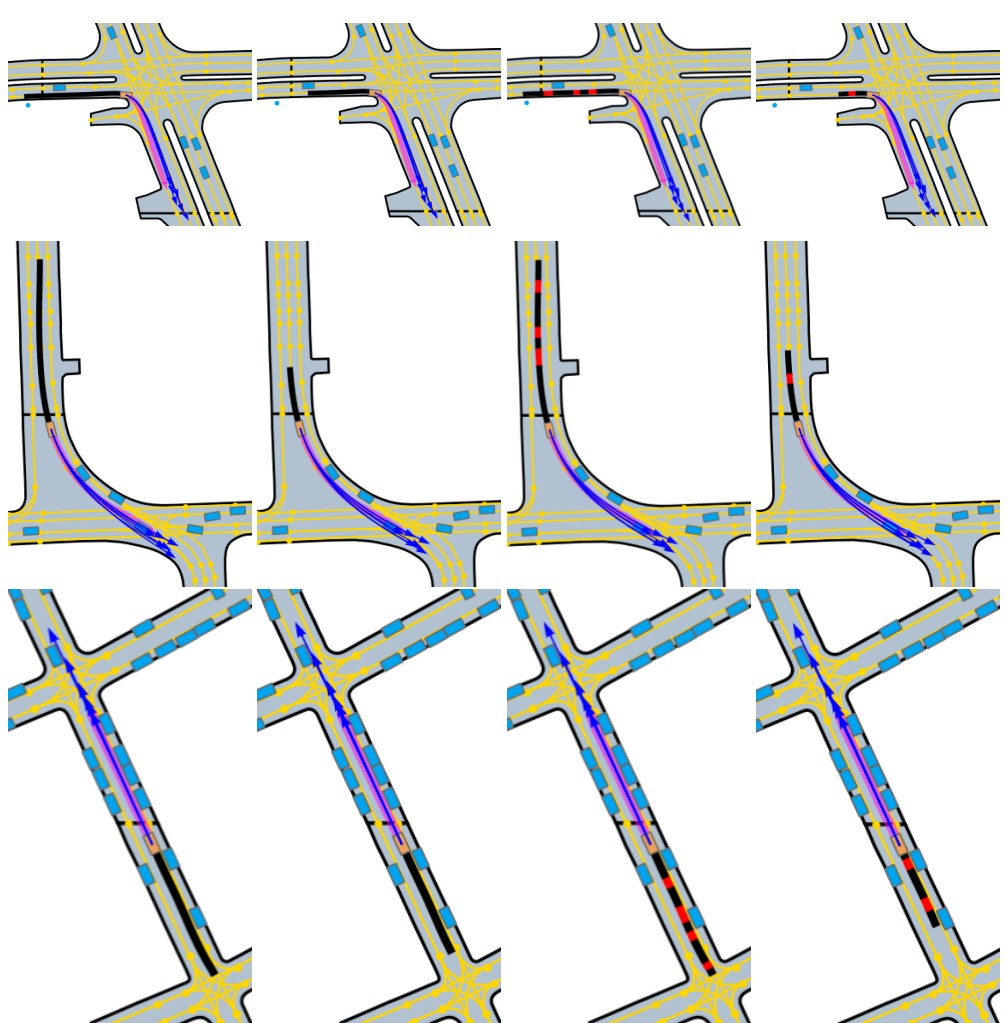

Figure 4: Qualitative results of the proposed model with varying input observations on Argoverse 2 dataset. Black: observed history; red: missing points; magenta: ground truth future; blue: predicted future trajectory.

