# OpenReview forum: "UIT-Pred: Universal Intermittent Trajectory Predictor for Autonomous Driving"
_ICLR.cc/2026/Conference — ICLR 2026 Conference Withdrawn Submission_

### Official Review · Reviewer_RJsH · 2025-10-25

**Soundness:** 2
**Presentation:** 2
**Contribution:** 2
**Rating:** 4
**Confidence:** 5

**Summary:**

This paper proposes UIT-Pred, a unified trajectory prediction framework for autonomous driving that effectively handles intermittent and variable-length input observations, such as those caused by occlusions, sensor limitations, or agents entering/exiting the scene. Its key contributions include: (1) a time-aware input representation using scaled timestamps and inter-observation intervals to model motion dynamics without fixed-length assumptions; (2) the Bidirectional Time-Decay Mamba (BTD-Mamba) module, which enhances state-space models with bidirectional processing and a temporal decay mechanism to maintain continuity across irregular observations; and (3) a novel prediction module with learnable state embeddings and cross-attention mechanisms to fuse agent history, multi-modal intentions, and map context. Extensive experiments on Argoverse 1 and 2 show that UIT-Pred achieves state-of-the-art performance across diverse and challenging input conditions.

**Strengths:**

1. Address the core points: The paper addresses two key challenges in real-world trajectory prediction - variable-length observations and data loss - which mainstream methods typically handle only one of or rely on complete data.
2. The structure of the paper is complete, including methods, experiments and visualization results.

**Weaknesses:**

1. **Novelty**: The overall architecture is highly similar to DeMo [1], with only minor modifications. Such incremental changes are insufficient to establish strong novelty for the paper.

2. **Experiment**: As shown in Table 1, the proposed method does not demonstrate significant improvements over DeMo across most metrics. Moreover, Table 2 does not specify whether the reported results on the Argoverse 1 and Argoverse 2 datasets are based on the validation split or the test split, which reduces the clarity and credibility of the experimental comparison.

3. **Figure**: The figures are not provided in vector format, leading to limited visual clarity and reduced readability in the manuscript.


**References**:

[1] DeMo: Decoupling Motion Forecasting into Directional Intentions and Dynamic States, NeurIPS 2024.

**Questions:**

See weaknesses.

---

> ### Author Response · Authors · 2025-12-04
>
> We thank the reviewer for their constructive feedback.
>
>
> **Weaknesses (W):**
>
>
> **W1**:
> UIT-Pred is not merely a minor modification of DeMo. While both methods perform trajectory prediction, they differ fundamentally in their motivations and problem settings, core architectural components, and experimental evidence.
>
> 1. **Motivations and Problem Settings:**
>    UIT-Pred solves a different problem that DeMo does not address. DeMo assumes a fully observed, fixed-length history with no missing points and no variable-length histories, whereas UIT-Pred provides a unified approach for handling all types of observations, including full, variable-length, missing-only, and mixed. This difference in problem formulation already creates a substantial novelty gap.
>
> 2. **Core Architectural Components:**
>    1. UIT-Pred introduces time-aware input features absent in DeMo, including scaled timestamps $t_i$ (Eq. 1) and inter-observation intervals $\Delta t_i$ (Eq. 2). This temporal encoding is a key novelty that enables handling intermittent observations in a single pipeline without explicit validity masks.
>    2. Unlike DeMo, which applies Mamba unidirectionally on complete observations, UIT-Pred’s BTD-Mamba handles intermittent observations bidirectionally and incorporates a learnable time-decay function, reflecting the real-world principle that older or delayed measurements should have reduced influence, a capability absent in DeMo.
>    3. UIT-Pred's Predictor Module introduces missing-step and variable-step temporal embeddings, enabling reconstruction of missing observations and unobserved variable-length segments. In contrast, DeMo predicts only future trajectories and mode probabilities. This represents a functional, not merely architectural, novelty.
>
> 3. **Experimental Evidence:**
>    1. UIT-Pred addresses multiple input scenarios that DeMo is not designed for (Table 1). In contrast, *DeMo-Orig* experiences significant performance drops across various input settings for minADE$_1$: Missing+Var (26.71%), Var. Obs (22.35%), Missing-Only (5.58%), and Full Obs (3.32%). UIT-Pred, however, remains stable and consistently superior. This highlights that UIT-Pred’s design is architecturally essential for the target domain; if it were merely a minor modification, it would not outperform DeMo so markedly under intermittent observations.
>
>
>
> **W2**: 1. Table 1 shows that UIT-Pred achieves the largest improvements over DeMo and DeMo-RSD across all input scenarios. Under full observations, UIT-Pred outperforms DeMo-RSD on minADE$_1$, minFDE$_1$, and MR$_1$ by 5.10%, 5.52%, and 4.01%, respectively.
>
> 2. In the updated paper, we revised Table 2 to explicitly specify that all Argoverse-1 and Argoverse-2 results are reported on the official validation split to ensure a fair comparison.
>
>
> **W3**: All figures in the revised PDF are now in vector formats (PDF/SVG) with uniform fonts and improved contrast for better readability.

---

### Official Review · Reviewer_sW8a · 2025-10-27

**Soundness:** 2
**Presentation:** 2
**Contribution:** 2
**Rating:** 4
**Confidence:** 3

**Summary:**

This paper proposes Universal Intermittent Trajectory Predictor, a universal trajectory prediction framework for autonomous driving that simultaneously handles variable-length historical trajectories and missing observations—common real-world challenges caused by sensor occlusions, communication delays, and temporal sparsity. The authors introduce three key components: (1) a time-aware input representation using timestamps and inter-observation intervals; (2) a Bidirectional Time Decay Mamba module that extends the Mamba architecture to capture forward and backward temporal dependencies while mitigating information gaps from missing data; and (3) a unified prediction module that encodes motion states and models multimodal future distributions. Extensive experiments on Argoverse 1 and Argoverse 2 demonstrate consistent performance gains across diverse observation conditions.

**Strengths:**

1. The joint modeling of variable-length inputs and missing data aligns closely with practical autonomous driving scenarios, addressing a notable gap in existing methods that assume fixed-length, complete trajectories.
2. Comprehensive experiments: The evaluation covers four realistic input conditions and compares against strong baselines (e.g., LaKD, RSD, DTO).
3. The paper includes detailed component-wise analyses (TAIR, BTD-Mamba, prediction module) and auxiliary loss ablations, validating the contribution of each design choice.

**Weaknesses:**

1. The model integrates BTD-Mamba, Transformers, GRUs, cross-attention, and multiple auxiliary tasks, raising concerns about computational cost and real-time deployability, critical for autonomous systems, but these aspects are not discussed.
2. While the method outperforms variants of HiVT and QCNet, it lacks comparison with cutting-edge approaches such as diffusion-based predictors or hybrid SSM-Transformer models under the same intermittent observation settings.
3. The use of a negative exponential decay (Eq. 5) is intuitive but not rigorously motivated or compared against alternatives (e.g., learnable gates, linear decay), leaving room for doubt about its optimality.
4. Missing observations are generated via random drop (RSD), which may not reflect real-world structured missingness (e.g., prolonged occlusions behind large vehicles), limiting the practical validation of robustness.

**Questions:**

1. Can the authors provide inference speed (FPS) or model size comparisons? Is UIT-Pred feasible for real-time deployment on embedded automotive platforms?
2. Does the bidirectional processing in BTD-Mamba violate causality during inference? If used online, how is the backward pass implemented without future observations?
3. Have the authors considered modeling missingness as a function of scene context (e.g., occlusion by nearby agents or map geometry), rather than relying solely on time intervals?
4. Is the model trained on mixed data (full + missing + variable-length), or can it generalize to missing scenarios when trained only on complete trajectories?

---

> ### Author Response · Authors · 2025-12-04
>
> We appreciate the reviewer’s insightful comments.
>
>
> **Weaknesses (W):**
>
> **W1**:  Although UIT-Pred integrates several components, the overall architecture is lightweight and designed for real-time operation.
> 1. UIT-Pred is designed for intermittent observation settings, achieving lower computational load by handling multiple input types within a single framework.
>
> 2. Auxiliary tasks operate only during training and add no inference cost.
>
> 3. We have added the runtime complexity analysis in the updated version of the paper, specifically in Section 4.2 and Table 8, for both datasets. UIT-Pred is highly computationally efficient across both Argoverse datasets. Training remains lightweight (7.53-7.48M parameters, 900-1200 min on two RTX A5000 GPUs), while inference uses only 6.55-6.50M parameters. UIT-Pred runs in real time, achieving 2.1–2.4 ms-per sample with just 0.51–0.52 GFLOPs.
>
> **Table:** Computational efficiency of the proposed model using two NVIDIA RTX A5000 GPUs.
> *Abbr.: TTT - Total Training Time, IT/S - Inference Time per Sample, TP - Training Parameters, IP - Inference Parameters, BS - Batch Size, F/S - FLOPs per Sample.*
>
> | Datasets       | TTT (min) | IT/S (ms) | TP (M) | IP (M) | BS  | F/S (G) |
> |----------------|-----------|-----------|--------|--------|-----|----------|
> | Argoverse 2    | 900       | 2.11      | 7.53   | 6.55   | 128 | 0.52     |
> | Argoverse 1    | 1200      | 2.45      | 7.48   | 6.50   | 128 | 0.51     |
>
>
> **W2**: Our work targets intermittent observation forecasting, a setting for which, to our knowledge, no prior method, including diffusion-based predictors or hybrid SSM-Transformer architectures, currently provides published implementations or evaluation results. Including these methods would require inventing an entire imputational and temporal interface for each model, which we believe would not be a fair comparison.
>
> **W3**: We evaluate the model under several alternative temporal-decay parameterizations, exponential decay, sigmoid gating, linear-clipped decay, and softplus-inverse decay, as summarized in Table 6 and section 4.2 of the revised paper’s appendix.
>
>
>
> **W4**: To reflect real-world structured missingness, we used block occlusions, gradient-guided removal, burst drops, and intersection-based occlusions. The corresponding results and analysis are provided in Tables 3 and 4 (Section 4.2) and Tables 10-11 (Appendix A.2.2 and A.2.3) in the revised paper.
>
> **Questions (Q):**
>
> **Q1**: Please refer to **W1**.
>
>
> **Q2**: BTD-Mamba as described does not require future observations and therefore does not violate causality. The “bidirectional” processing runs over the observed history. Both directions only use data up to the current observation time, so the module is causal at inference.
>
> **Q3**: This is a valuable idea and aligns well with potential future extensions of UIT-Pred.
>
>
> **Q4**: Our default configuration trains on a mixture of complete trajectories and trajectories with simulated intermittent observations (variable-length truncation & random drop).
>
> We also evaluate a model trained only on complete trajectories across four inference regimes: missing + variable-length input, variable-length input, missing-only input, and fully observed sequences. The results are presented in Table 1, Section 4.2 of the revised paper.
>
> UIT-Pred (trained on complete trajectories; *Train-full*) maintains strong performance with only marginal degradation relative to mixed-trained models, indicating that its robustness stems primarily from the architecture itself.

---

### Official Review · Reviewer_UY3o · 2025-11-01

**Soundness:** 3
**Presentation:** 3
**Contribution:** 3
**Rating:** 6
**Confidence:** 3

**Summary:**

This paper introduces UIT-Pred, a unified architecture for trajectory prediction in autonomous driving that robustly handles variable-length input histories and missing data. The method combines a time-aware input representation, a Bidirectional Time Decay Mamba (BTD-Mamba) module (extending state space models with bidirectional and decay-aware processing), and a multimodal prediction module using integrated state encoding and cross-attention fusion. The approach is validated empirically on the Argoverse 1 and Argoverse 2 benchmarks, demonstrating state-of-the-art performance under several observation regimes and scenarios. Extensive ablations and qualitative results further support the claims.

**Strengths:**

+ The paper tackles an important and under-addressed real-world challenge: predicting agent trajectories under intermittent (i.e., variable-length and missing) observation sequences, directly motivated by realistic issues such as sensor occlusion and communication drops in autonomous driving.
+ The method brings together several principled ideas—temporal normalization, bidirectional SSMs with time decay, and unified fusion—to handle variable and incomplete data without reliance on input masking, explicit validity masks, or duplicative augmentation. The proposed Bidirectional Time Decay Mamba module is theoretically motivated and technically well-integrated, extending recent progress in SSMs in sequence modeling to the irregular sampling domain.

**Weaknesses:**

1. While the decay process in BTD-Mamba is heuristically motivated (see Equations 4–5), the formulation and parameterization of the decay via an MLP, as well as the use of a negative exponential, seem ad-hoc. There is no theoretical analysis on how this decay mechanism affects information retention or gradient propagation through highly irregular gaps. A more principled or formal treatment would be valuable.
2. Ambiguity in Loss Weighting and Training Objective: The main loss function (Equation 23 and detailed appendix losses) weights all terms equally without analysis or justification. No tuning or sensitivity study is reported regarding these hyperparameters. In multi-loss systems, uniform weighting is rarely optimal. This could materially affect empirical results (see Equations 24–33, Table 8 in the appendix for auxiliary losses – but no broader tuning study).
3. Certain key steps gloss over important mathematical details: the cross-attention decoders used extensively for graph, lane, and multimodal fusion are described abstractly ($\mathcal{D}(·)$ in Equations 14–20) but lack explicit architectural detail (are positional encodings used? What are the dimensions of queries/keys/values in multi-modal context?).
4.  There are a few spots where notational clarity is lacking, particularly in not distinguishing between predicted, reconstructed, and observed trajectories within tensors in the main methodology (Equations 15–17). Indexing switches between $j$, $t$, and $l$ within equations can be confusing.

**Questions:**

1. How robust is UIT-Pred to adversarial missing patterns? Any chance for synthetic stress-testing to be added or analyzed?
2. In the fusion module, are positional encodings used within the cross-attention layers (cf. Equations 14-20)? If so, how are these adapted for variable-length or irregularly timed data?
3. Why did the authors choose to weight all loss terms equally in $L_\text{total}$? (cf. equation 23)

---

> ### Author Response · Authors · 2025-12-04
>
> Thank you to the reviewer for this constructive feedback.
>
> **Weaknesses (W):**
>
> **W1**:To validate the necessity of the learned decay formulation in BTD-Mamba, we conducted ablations with:
> (i) **a fixed exponential decay** (using a constant value of 0.5 instead of the MLP),
> (ii) **no decay,** and
> (iii) **learned exponential decay**. The updated results and analysis are provided in Section 4.2 and Table 7 of the revised paper.
>
> Both fixed-decay and no-decay variants degrade noticeably under intermittent observations, while UIT-Pred’s learned decay maintains superior performance. Fixed decay fails because a single rate cannot accommodate diverse motion or gap lengths, and no-decay performs worst as stale states accumulate across long gaps. These results highlight that adaptive, learned decay is crucial for robustness in irregular observation settings.
>
> **Table:** Impact of fixed, no, and learned decay in BTD-Mamba for the Argoverse 2 validation set.
>
> | Method         | minADE$_1$ | minFDE$_1$ | MR$_1$ | minADE$_6$ | minFDE$_6$  | MR$_6$  |
> |----------------|---------------|---------------|----------|---------------|---------------|----------|
> | Fixed-decay    | 1.6824        | 4.1138        | 0.6036   | 0.7045        | 1.3459        | 0.1858   |
> | No-decay       | 1.6529        | 4.0284        | 0.5903   | 0.6925        | 1.3163        | 0.1628   |
> | Learned-decay  | 1.5882        | 3.9402        | 0.5717   | 0.6562        | 1.2326        | 0.1551   |
>
>
> Additionally, the revised paper includes an ablation study of various decay mechanisms such as sigmoid gating, linear-clipped decay, and softplus-inverse decay, summarized in Table 6 and discussed in Section 4.2.
>
>
>
> **W2**: We have added a loss-weight sensitivity analysis in the revised paper, presented in Appendix A.2.7 (Table 16), evaluating the impact of different weighting schemes on model performance on the Argoverse 2 dataset.
>
> We compare:
> 1. **Fixed weighting schedules** (first row)
> 2. **Learned uncertainty-based weighting** (second row)
> 3. **Uniform weighting** (third row)
>
> Notably, the uniform weighting configuration $w_{reg}$ = $w_{cls}$ = $w_u$ = $w_m$ = $w_e$ = 1 achieves the best overall performance, with the lowest minADE, minFDE, and MR across both top-1 and top-6 predictions.
>
> This suggests that while uncertainty-based weighting can adaptively balance tasks, equal weighting is sufficient in our setup, providing strong performance without additional complexity. The results highlight the robustness of our multi-task training formulation and indicate that all task-specific losses contribute meaningfully to trajectory prediction.
>
> **Table:** Weight sensitivity study of loss for Argoverse 2 validation dataset.
>
> | $w_{reg}$ | $w_{cls}$ | $w_u$ | $w_m$ | $w_e$ | minADE$_1$ | minFDE$_1$  | MR$_1$  | minADE$_6$  | minFDE$_6$  | MR$_6$ |
> |-------------------|-------------------|---------|---------|---------|---------------|---------------|----------|---------------|---------------|----------|
> | 1                 | 1                 | 0.5     | 0.5     | 0.2     | 1.6204        | 4.0284        | 0.5723   | 0.6569        | 1.2664        | 0.1554   |
> | -                 | -                 | -       | -       | -       | 1.6214        | 4.1926        | 0.6098   | 0.6783        | 1.2616        | 0.1722   |
> | 1                 | 1                 | 1       | 1       | 1       | 1.5882        | 3.9402        | 0.5717   | 0.6562        | 1.2326        | 0.1551   |
>
>
> **W3**: UIT-Pred intentionally omits positional encodings, as fixed index-based positional schemes (absolute or relative) are not compatible with the intermittent observation patterns the model is designed to handle. Instead, the Time-Aware Input Representation (TAIR) already provides the requisite temporal signal, rendering additional positional embeddings unnecessary for accurate trajectory prediction. A standard Transformer decoder $\mathcal{D}(\cdot, \cdot, \cdot)$ is used for cross-attention, where the first argument is the query, the second is the key, and the third is the value. Moreover, we have updated the paper to include the dimensions of the queries, keys, and values in Equations 14–20.
>
> **W4**: We have addressed the notational ambiguities in the revised version of the paper. Specifically, we have clarified the distinction between predicted, reconstructed, and observed trajectories in Equations 15-17 and standardized the indexing across tensors for consistency.
>
> **Questions (Q):**
>
> **Q1**: To validate robustness, we conducted synthetic stress tests covering block occlusions, gradient-guided removal, burst drops, and intersection-based occlusions. The corresponding results and analysis are provided in Tables 3 and 4 (Section 4.2) and Tables 10-11 (Appendix A.2.2 and A.2.3) in the revised paper.
>
> **Q2**: Please refer **W3**.
>
> **Q3**: Please refer **W2**.

---

### Official Review · Reviewer_cpBU · 2025-11-01

**Soundness:** 3
**Presentation:** 2
**Contribution:** 2
**Rating:** 4
**Confidence:** 4

**Summary:**

This paper proposes UIT-Pred, a trajectory prediction framework for autonomous driving that is designed to work under intermittent observation, e.g. variable-length histories and missing inputs. It uses time-aware input representation so the model doesn’t need to rely on fixed windows or masks. It uses a Bidirectional Time Decay Mamba (BTD-Mamba) to capture sequential dependencies in both forward and backward directions across input observations. Experiments on Argoverse 1 and 2 Datasets show consistent gains over baselines (HiVT, QCNet, DeMo, Forecast-MAE).

**Strengths:**

1. The time-aware input representation allows the model reason about intermittency without validity masks.
2. It extends Mamba to BTD-Mamba to preserve temporal coherence under missing observations.
3. On Argoverse 1 and 2 Datasets, the model outperforms mask-based baselines.

**Weaknesses:**

1. The notations in this paper are very confusing. Usually we use $x_t$ for position at time $t$, but this paper uses $t_x$ for position and $t_{vel}$ for velocity. The paper also uses $t_i$ for timestamps, $\Delta t_i$ for gaps, and $t’, t’’$ for temporal embeddings.
2. The core ideas of time-aware input features (scaled timestamps, inter-observation gaps) and a Bidirectional Time-Decay Mamba are reasonable, but read as modest adaptations of known ingredients (temporal features + bidirectional/decay in SSMs) rather than a clearly new idea.
3. The baselines this paper is comparing against are old. For example, on the current Argoverse 2 leaderboard, top models can achieve MR$_6$ of 0.11 or 0.12, and minADE$_1$ of 1.42.
4. No complexity/runtime is given.

**Questions:**

There are many clear typos and small grammar issues. Here is a list:
1. Line 21: “Particularly, We extend” → “Particularly, we extend”
2. Line 40: “are are designed” → “are designed”
3. Line 41: “observations are often intermitten” → intermittent
4. Line 61: “variale-length” → “variable-length”
5. Line 72-73: “intervals feature” → “interval features”
6. Line 85: “to effectively captures” → “to effectively capture”
7. Line 107: “pedestrians, or cyclist” → “pedestrians, or cyclists”
8. Line 420: “Observations Lengths” → “Observation Lengths”

---

> ### Author Response · Authors · 2025-12-04
>
> We appreciate the reviewer’s suggestion.
>
> **Weaknesses (W):**
>
> **W1**: We have revised the notation for clarity in the updated version of the paper: coordinates are changed from $t_x^{(i)}$ to $x_t^{(i)}$, and velocity from $t_{vel}^{(i)}$ to $vel_{t}^{(i)}$.
>
>
> **W2**: 1. **Novelty of Time-Aware Input Representation (TAIR):**
>    By integrating scaled timestamps and inter-observation intervals into the input representation, our approach yields a single, unified input format that is inherently mask-free and augmentation-free. This formulation applies seamlessly across all intermittent-observation settings, including variable-length, missing-only, mixed, and fully observed sequences, without requiring separate input pipelines or handcrafted preprocessing. To our knowledge, no prior trajectory-prediction method in autonomous driving supports this level of generality or converts heterogeneous observation patterns into a single consistent representation.
>
> 2. **Novelty of Bidirectional Time-Decay Mamba (BTD-Mamba):**
>    To our knowledge, no prior SSM model incorporates time-dependent decay in both forward and backward directions to handle intermittent observations in autonomous-driving trajectory prediction. Our learned decay terms are explicitly parameterized by real inter-arrival gaps, allowing the recurrent state to remain temporally aware of missing intervals, capabilities absent in standard Mamba and its variants.
>
>
> **W3**: Leaderboard SOTA numbers (minADE ≈ 1.42, MR ≈ 0.11) are achieved using full, dense histories; however, these models cannot operate under intermittent observation settings without substantial modification. Therefore, we compare against baselines specifically designed for intermittent observations, such as Forecast-mae-RSD and DeMo-RSD in Table 1 and Table 9. For varying observation lengths, we compare with LaKD, DTO, FLN, HiVT-RM, and QCNet-RM in Table 2.
>
>
> **W4**: We have added the runtime complexity analysis in the updated version of the paper, specifically in Section 4.2 and Table 8, for both datasets. UIT-Pred is highly computationally efficient across both Argoverse datasets. Training remains lightweight (7.53–7.48M parameters, 900–1200 min on two RTX A5000 GPUs), while inference uses only 6.55–6.50M parameters. UIT-Pred runs in real time, achieving 2.1–2.4 ms per sample with just 0.51–0.52 GFLOPs.
>
> **Table:** Computational efficiency of the proposed model using two NVIDIA RTX A5000 GPUs.
> *Abbr.: TTT - Total Training Time, IT/S - Inference Time per Sample, TP - Training Parameters, IP - Inference Parameters, BS - Batch Size, F/S - FLOPs per Sample.*
>
> | Datasets       | TTT (min) | IT/S (ms) | TP (M) | IP (M) | BS  | F/S (G) |
> |----------------|-----------|-----------|--------|--------|-----|----------|
> | Argoverse 2    | 900       | 2.11      | 7.53   | 6.55   | 128 | 0.52     |
> | Argoverse 1    | 1200      | 2.45      | 7.48   | 6.50   | 128 | 0.51     |
>
> **Questions**:  We have revised the updated paper to correct typos and minor grammatical issues based on your suggestions.

---

### Note · Authors · 2026-03-17

I have read and agree with the venue's withdrawal policy on behalf of myself and my co-authors.

---

### Meta-Review · Area_Chair_jvtF · 2026-01-10

**Summary:**

Some reviewers find that the contributions of the paper are primarily an adaptation of existing methods rather than significant innovations: "The overall architecture is highly similar to DeMo, with only minor modifications." Also, there are concerns that the baselines used in the experiments are outdated, so it is hard to evaluate the performance of the proposed method compared to state-of-the-art approaches. In addition, it is not clear how the model performs under adversarial missing patterns and whether it can generalize well to real-world scenarios (real-world structured missingness). There are many typos, as suggested by the reviewers.

**Reviewer Concerns:**

The authors have provided rebuttals to reviewers' concerns. The authors have clarified that the problem solved by DeMo is different from the proposed method, as DeMo assumes fully-observed trajectories. The authors also explain that leading methods on the benchmark do not work with intermittent observation settings. However, the AC still feels that the technical contribution is limited, given that the method's architecture is similar to DeMo. Another weakness of the work is how to simular to missing intermittent observations, like random drop, which do not look realistic in real-world applications, and the authors can look into a better simulation. The AC does not recommend acceptance of the work to ICLR, given the presented concerns.

**Reviewer Scores:**

The reviewers of this paper gave ratings of 4,4,4,6. I don't think the scores would be changed.

---

### Decision · Program_Chairs · 2026-01-26

Reject